

# Atmospheric photooxidation and ozonolysis of sabinene: Reaction rate constants, product yields and chemical budget of radicals

Jacky Y. S. Pang[1], Florian Berg[1], Anna Novelli[1], Birger Bohn[1], Michelle Färber[1], Philip T. M. Carlsson[1], René Dubus[1], Georgios I. Gkatzelis[1], Franz Rohrer[1], Sergej Wedel[1], Andreas Wahner[1], and Hendrik Fuchs[1,2]

[1]Institute of Energy and Climate Research, IEK-8: Troposphere, Forschungszentrum Jülich GmbH, Jülich, Germany
[2]Department of Physics, University of Cologne, Cologne, Germany

*Correspondence to*: Hendrik Fuchs (h.fuchs@fz-juelich.de)

**Abstract.** The oxidation of sabinene by the hydroxyl radical (OH) and ozone ($O_3$) was investigated under atmospherically relevant conditions in the atmospheric simulation chamber SAPHIR at Forschungszentrum Jülich, Germany. The rate constants of the reactions of sabinene with OH and with $O_3$ were determined. The temperature dependence between 284 K to 340 K of the rate constant of the reaction of sabinene with OH, $k_{SAB+OH}$, was measured for the first time using an OH reactivity instrument, resulting in an Arrhenius expression of $(1.67\pm0.16)\times10^{-11}\times\exp((575\pm30)T^{-1})$ cm$^3$ s$^{-1}$. The values agree with those determined in chamber experiments in this work and reported in the literature for ~298 K within the uncertainties of measurements. The ozonolysis reaction rate constant of sabinene ($k_{SAB+O3}$) determined in chamber experiments at a temperature of $(278\pm2)$ K is $(3.7\pm0.5) \times 10^{-17}$ cm$^3$ s$^{-1}$, which is 55 % lower than the value reported in the literature for room temperature. The measurement of products from the oxidation of sabinene by OH resulted in an acetone yield of $(21\pm15)$ %, a formaldehyde yield of $(46\pm25)$ %, and a sabinaketone yield of $(19\pm16)$ %. All yields determined in the chamber experiments agree well with values from previous laboratory studies within their uncertainties. In addition, the formaldehyde yield determined in this study is consistent with that predicted by the sabinene OH-oxidation mechanism which was devised from quantum chemical calculations by Wang and Wang (2018), whereas the acetone yield is about 15 % absolute higher than that predicted by the mechanism. In the ozonolysis experiments, the analysis if product measurements results in an acetone yield of $(5\pm2)$ %, a formaldehyde yield of $(48\pm15)$ %, a sabinaketone yield of $(31\pm15)$ %, and an OH radical yield of $(18\pm25)$ %. The OH radical yield is lower than expected from the theoretical mechanism in Wang and Wang (2017), but the value still agrees within the uncertainty. An analysis of the chemical budget of OH radicals was performed for the chamber experiments. The analysis reveals that the destruction rate of OH radcials matches the production rate of OH suggesting that there is no significant missing OH source for example from isomerization reactions of peroxy radicals for the experimental conditions in this work.

## 1 Introduction

Monoterpenes are an important constituent of biogenic volatile organic compounds (BVOCs). About 160 Tg of monoterpenes are released into the atmosphere each year (Guenther et al., 2012). They play an important role in tropospheric chemistry and



the formation of secondary pollutions such as ozone ($O_3$) and particles due to their high reactivity toward major oxidants in the atmosphere that include hydroxyl radicals (OH), $O_3$ and nitrate radicals ($NO_3$) (Atkinson and Arey, 2003). Sabinene contributes up to 7 % to the total monoterpene emissions and is the 5th-most abundant monoterpene species in the atmosphere (Sindelarova et al., 2014). Sabinene is for example emitted by beech (Tollsten and Müller, 1996), birch (Hakola et al., 1998),

and oak trees (Staudt and Bertin, 1998), making sabinene the major monoterpene emission in some European (Hakola et al., 2003; Holzke et al., 2006) and Asian forests (Kim et al., 2005).

The chemical structure of sabinene is similar to that of β-pinene, a monoterpene with an exocyclic C-C double bond and bicyclic rings. The oxidation of β-pinene has been extensively investigated in many laboratory (e.g., Kaminski et al., 2017; Xu et al., 2019) and theoretical studies (e.g., Nguyen et al., 2009; Vereecken and Peeters, 2012), while only few studies have been

performed with sabinene. Sabinene differs from β-pinene by having a three-membered bridging ring instead of a four-membered ring. The lower number of carbon atoms in the bicyclic ring of sabinene leads to a higher strain in the ring impacting the reaction pathways in the oxidation of sabinene.

Organic peroxy radicals ($RO_2$) are formed upon the oxidation by OH radicals and $O_3$. The subsequent chemistry of the $RO_2$ radicals depends mostly on the availability of nitric oxide (NO) in the atmosphere as NO rapidly reacts with $RO_2$ radicals, so

that other reaction pathways often cannot compete. In addition to bi-molecular reactions, some $RO_2$ radicals can undergo unimolecular reactions that are competitive with bi-molecular reactions in environments with a low NO mixing ratios (< 1 ppbv) such as forests. The position of functional groups in the $RO_2$ radical affects the rate of unimolecular reactions (Vereecken and Nozière, 2020). Unimolecular $RO_2$ reactions can lead to the regeneration of OH radicals and therefore enhance the self-cleansing ability of the atmosphere. This is for example known for $RO_2$ radicals produced from isoprene and methacrolein

(e.g., da Silva et al., 2010; Crounse et al., 2011, 2012; Fuchs et al., 2013, 2014; Peeters et al., 2014; Novelli et al., 2020), which partly explains the deficit in the OH production rate found in field experiments in isoprene-rich environments at low NO concentrations (Lelieveld et al., 2008; Whalley et al., 2011). The exact fate of $RO_2$ radicals in their subsequent chemistry determines the distribution of oxidation products and the yield of secondary organic aerosol.

There are only a few studies investigating specifically the chemical budget of radicals in the oxidation chain of monoterpenes.

The available literature indicates that current chemical models underestimate the formation of hydroperoxyl radicals ($HO_2$) in the photooxidation of α-pinene and β-pinene (Kaminski et al., 2017; Rolletter et al., 2019). There has been no study investigating the chemical budget of radicals in the oxidation of sabinene.

In this study, the oxidation of sabinene by OH and $O_3$ was investigated in experiments in the atmospheric simulation chamber SAPHIR (Simulation of Atmospheric Photochemistry In a Large Reaction Chamber) at Forschungszentrum Jülich, Germany.

Experiments were performed at ambient conditions with the aim to improve the understanding of the oxidation mechanism of sabinene, which included the determination of reaction rate constants with oxidants (OH and $O_3$), product yields (acetone, formaldehyde, and OH from the ozonolysis reaction) and the chemical budget of OH radicals. Additional measurements were





performed in the laboratory to determine the temperature dependence of the rate constant of the reaction of sabinene with OH radicals.

## 2 Oxidation mechanism of sabinene

There are only a few experimental and theoretical studies on the oxidation mechanism of sabinene and to the best of our knowledge a detailed oxidation mechanism of sabinene including the subsequent chemistry of oxidation products is not available (Carrasco et al., 2006; Wang and Wang, 2017, 2018; Almatarneh et al., 2019). Figure 1 and Figure 2 show the current knowledge of the oxidation mechanisms of sabinene by OH (Wang and Wang, 2018) and $O_3$ (Wang and Wang, 2017), derived from quantum chemical calculations.

In the reaction with OH, sabinene either undergoes hydrogen abstraction or OH addition at the exocyclic C-C double bond. OH addition is predicted to be the dominant pathway from structure-activity relationships (SAR) (Peeters et al., 2007) and quantum chemical calculations (Wang and Wang, 2018). Several alkyl radical isomers can be produced from the H-abstraction reaction, but their total yield is only 4 % to 8 % (reaction path (c) in Figure 1). OH-addition to the cyclic carbon results in a primary alkyl radical with a yield of 47 % (reaction path (a) in Figure 1) and addition to the terminal carbon results in a tertiary alkyl radical with a yield of 45 % (reaction path (b) in Figure 1). Yields are derived from the energy barrier of the OH addition reactions in the quantum chemical calculations by Wang and Wang (2018). It is worth noting that the OH-addition at the cyclic carbon is much more favorable in sabinene than in β-pinene, for which the yield of the primary alkyl radical from the OH addition to the cyclic carbon atom is only 8 % and the yield of the tertiary alkyl radical from the OH addition to the terminal carbon atom is 92 %.

After the addition of OH, the tertiary alkyl radical quickly isomerizes by breaking the three-membered ring resulting in a $RO_2$ radical (SABINOHBO2, Figure 1) after the reaction with oxygen molecules ($O_2$). The ring-opening reaction is faster than the immediate addition of $O_2$ under atmospheric conditions resulting in a yield of 99 % of $RO_2$ radical SABINOHBO2 (Wang and Wang, 2018). The impact of strain in the 3-membered ring is apparent when comparing the yield of the ring-opening $RO_2$ radical SABINOHBO2 to the analogous ring-opening $RO_2$ radical from the OH-oxidation of β-pinene that has a 4-membered ring. In the case of β-pinene, ring-opening is less competitive than the immediate $O_2$ addition and the yield of the ring-opening $RO_2$ radical is only 30 % (Vereecken and Peeters, 2012).

The $RO_2$ radical SABINOHBO2 can either further react in bi-molecular reactions like other organic $RO_2$ radicals (e.g., with NO) forming eventually a hydroxyketone (sum formula: $C_7H_{10}O_2$) and acetone. According to quantum chemical calculations in Wang and Wang (2018), SABINOHBO2 can also undergo a unimolecular reaction with a rate constant of $k \sim 5$ s$^{-1}$ that eventually leads to the formation of a stable oxidation product with the sum formula $C_{10}H_{16}O_5$ containing two hydroperoxide groups. The predicted reaction rate constant makes this unimolecular reaction competitive with bi-molecular reactions even in the presence of a high NO mixing ratio (~20 ppbv).



The primary alkyl radical from the addition of OH to the cyclic carbon (reaction path (a) in Figure 1) forms a $RO_2$ radical
SABINOHAO2, which mainly reacts with NO, $HO_2$, and $RO_2$ in bi-molecular reactions. Rate constants of unimolecular
reactions are predicted to be too slow ($k < 10^{-3}$ $s^{-1}$, Wang and Wang (2018)) to be competitive with bi-molecular reactions. The
reaction between SABINOHABO2 and NO forms eventually a $HO_2$ radical together with sabinaketone (sum formula: $C_9H_{14}O$).

Following the theoretical study by Wang and Wang (2017), the ozonolysis of sabinene results in the production of either
sabinaketone together with formaldehyde oxide ($CH_2OO$) with a yield of 17 % (reaction path (A) in Figure 2) or formaldehyde
(HCHO) together with either one of two types of Criegee intermediates (CI-1 with a yield of 45 % through reaction pathway
(C), or CI-2 with a yield of 38 % through reaction pathway (B), Figure 2). The respective yields are similar to the analogous
pathways in the ozonolysis of β-pinene having yields of 5 % for $CH_2OO$, and 46 % and 49 % for the analogous Criegee
intermediates CI-1 and CI-2, respectively (Nguyen et al., 2009). In the atmosphere, formaldehyde oxide ($CH_2OO$) reacts
mainly with water to form formic acid and hydroxymethyl hydroperoxide (HMHP) (Long et al., 2016; Nguyen et al., 2016;
Vereecken et al., 2017). The Criegee intermediate CI-2 also reacts mainly with water to form α-hydroxyalkyl hydroperoxide
(AHAP, Figure 2) under humid conditions, whereas in dry conditions a competitive unimolecular reaction leads to the
formation of lactones. The α-hydroxyalkyl hydroperoxide can further decompose to sabinaketone in a reaction that is catalyzed
by water and acids making the yield of sabinaketone depends on the water vapor concentration. This might explain the
difference in the sabinaketone yield expected from the branching ratio of the ozonolysis pathway (A) of 17 % in the mechanism
by Wang and Wang (2017) and yields between 35 % to 50 % observed in experiments performed in the presence of water
vapor. (Hakola et al., 1994; Yu et al., 1999; Chiappini et al., 2006). The Criegee intermediate CI-1 exclusively undergoes a 1-
4 H-shift reaction forming a vinyl hydroperoxide (VHP) that subsequently decomposes to an OH radical and a vinoxy radical.
The vinoxy radical then reacts with $O_2$ forming a $RO_2$ radical (SABINO3O2), which is expected to further react in bi-molecular
reactions with NO, $HO_2$, and other $RO_2$ radicals leading eventually to closed-shell oxidation products.




**Figure 1.** Simplified mechanism of the oxidation of sabinene by OH by Wang and Wang (2018). Only the pathways that are
relevant for the experimental conditions in this study are shown.





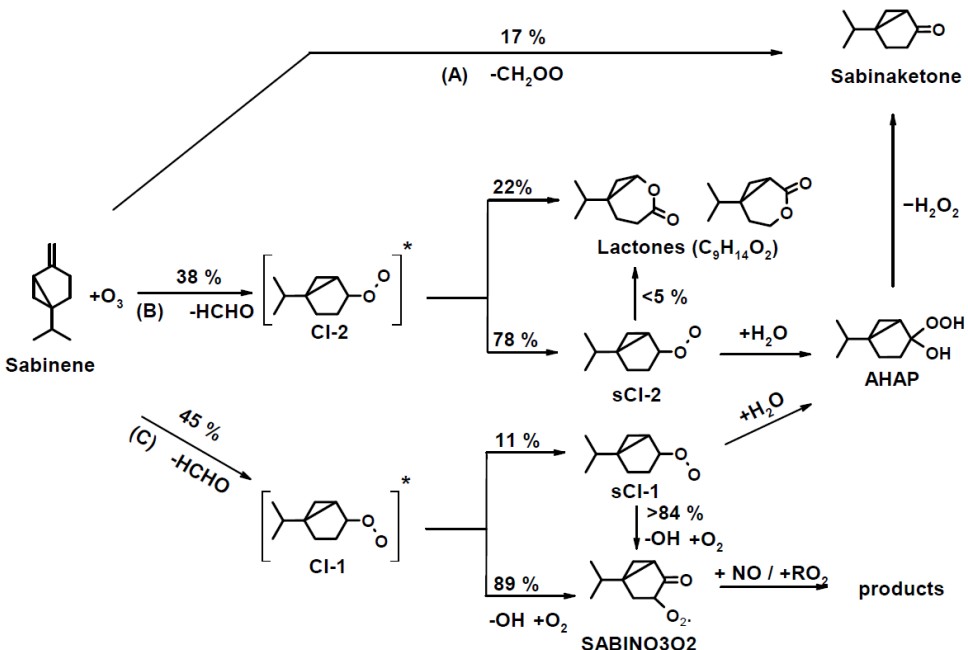


**Figure 2.** Simplified mechanism of the ozonolysis of sabinene by Wang and Wang (2017).

## 3 Methods

### 3.1 Determination of the temperature-dependence of $k_{SAB+OH}$ with OH reactivity measurements

The rate constant of the reaction between sabinene and OH at temperature between 284 K and 340 K were determined in the laboratory from OH reactivity ($k_{OH}$) measurements using a laser flash-photolysis laser-induced fluorescence (LP-LIF) instrument (Lou et al., 2010; Fuchs et al., 2017). In a temperature-controlled flow tube, OH radicals are generated in-situ by photolysis of $O_3$ using laser pulses of a quadrupled Nd:YAG laser at a wavelength of 266 nm and a low pulse repetition rate of 1 Hz. Air containing a well-known concentration of sabinene is continuously passed through the flow-tube. Sabinene reacts

with OH, which leads to the destruction of OH radicals. The decreasing OH radical concentration is detected via laser-induced fluorescence after excitation by short laser pulses (308 nm wavelength, 8.5 kHz pulse repetition frequency) in a low-pressure (~4 hPa) cell and monitored by single photon counting. A single exponential fit to the time series of the OH concentration directly gives the OH reactivity.



Gas mixtures of sabinene in synthetic air were prepared by injecting liquid sabinene (Roth Chemicals, GC grade, purity >98
%) with a syringe in an evacuated SilcoNert coated canister (Restek, volume 6 L). The canister was subsequently pressurized
to up to 3.5 bar with pure synthetic air prepared from ultrapure liquid nitrogen and oxygen (79 % $N_2$, 21 % $O_2$, Linde, purity
> 99.9999 %), resulting in mixing ratios of about 6 ppmv sabinene. The concentration of sabinene in the canister was
determined by measuring the total organic carbon (TOC) concentration using a catalytic oxidation at high temperature. In this
method, a small flow (500 sccm) from the canister was flowed over a palladium catalyst at a temperature of 760 °C. The
concentration of carbon dioxide was measured by a cavity ring-down spectrometer (CRDS, Picarro). Assuming that all carbon
stems from sabinene, its concentration in the canister can be calculated from the measured $CO_2$ concentrations.

In the OH reactivity instrument, small flows (10 sccm) of sabinene and of synthetic air (100 sccm), to which $O_3$ was added,
were mixed into a high flow (20,000 sccm) of humidified synthetic air (water mixing ratio between 0.7 and 1.3 %) in the flow
tube. The temperature of the flow tube was controlled by circulating water within a range of 10 °C to 70 °C and monitored
with two PT100 temperature sensors. All flows were controlled by calibrated mass flow controllers. The sabinene mixing ratio
in the flow was circa 5 ppbv. The $O_3$ concentration in the flow tube was 22 ppbv measured by an $O_3$ analyzer.

In total, three experiments were performed to measure the rate constants of the reaction of sabinene and OH ($k_{\mathrm{SAB+OH}}$) at seven
different temperatures between 10 °C to 70 °C. Two batches (A and B) of sabinene gas mixtures were measured, with batch
A being measured twice. After reaching a stable temperature, the OH reactivity of the air without sabinene (zero reactivity)
was first measured for about 30 minutes, followed by the measurement of air with sabinene for another 40 minutes. The
procedure was then repeated at different temperatures. The OH reactivity of air with sabinene was subtracted by its
corresponding zero reactivity and the rate constant of the OH reaction was calculated by using the sabinene concentration in
the canister ($[\mathrm{SAB}]_0$) and the dilution factor $f_{\mathrm{dil}}$ determined from the flow rates:

$$k_{\mathrm{SAB+OH}} = k_{\mathrm{OH}} \cdot ([\mathrm{SAB}]_0 \cdot f_{\mathrm{dil}})^{-1} \tag{1}$$

The determination of the sabinene concentration in the canister with the TOC method is the predominant contributor to the
uncertainty of the calculated reaction rate constants, resulting in uncertainties of about 2.5 % to 5.0 %. The loss of sabinene to
$O_3$ can be neglected (0.005 % at 20 °C) due to the short residence time of less than two seconds in the flow tube.

The temperature dependence of the reaction rate constant $k_{\mathrm{SAB+OH}}$ can be expressed by the Arrhenius equation:

$$k_{\mathrm{SAB+OH}}(T) = A \cdot \exp\left(-\frac{E_A}{R} \cdot \frac{1}{T}\right) \tag{2}$$

where $A$ is a pre-exponential factor, $E_A$ is the activation energy, $T$ is the temperature and $R$ is the universal gas constant. The
temperature dependence coefficient $-E_A R^{-1}$ is determined by a regression analysis of the reaction rate constant $k_{\mathrm{SAB+OH}}$ as a
function of the inverse temperature $T^{-1}$.



## 3.2 Atmospheric simulation chamber SAPHIR

Sabinene oxidation experiments were performed in the atmospheric simulation chamber SAPHIR. A detailed description of the chamber can be found in previous publications (e.g., Rohrer et al., 2005; Kaminski et al., 2017). In brief, SAPHIR is a cylindrically shaped (5 m diameter, 18 m length) outdoor chamber with an inner volume of 270 $m^3$ which is confined by a

double-walled Teflon film (FEP) The high volume-to-surface ratio minimizes wall effects. A shutter system can be opened and closed for experiments to be performed in the dark (shutters closed) or in the sunlit (shutters opened) chamber. The entire spectrum of solar radiation is transmitted by the FEP film allowing for photooxidation experiments to be performed under natural conditions. The air pressure inside the SAPHIR chamber is kept slightly above atmospheric pressure (15 Pa) to ensure that air outside the chamber cannot leak into the chamber. The replenishment flow required to keep the overpressure results in

the dilution of trace gases with a rate of approximately 4 % $h^{-1}$. The temperature in the chamber is similar to ambient temperature.

All experiments in this work were performed in synthetic air produced from evaporating ultrapure liquid oxygen and nitrogen (Linde, purity > 99.9999 %). Before the start of the experiment, the chamber was flushed with synthetic air until trace gas concentrations were below the detection limit of the instruments. The air was then humidified by flushing water vapor from

boiling Milli-Q water into the chamber together with a high flow of synthetic air. In the illuminated chamber, nitrous acid (HONO), NO, acetone, and HCHO are produced at a rate of several hundred pptv $hr^{-1}$ presumably from chamber wall reactions (Rohrer et al., 2005). Production rates were determined from the measured increase of their concentrations before sabinene was injected. The photolysis of HONO by sunlight is the major source of OH and NO in the experiments.

## 3.3 Instrumentation

The set of instruments used in the experiments in this work is listed in Table 1. The concentrations of $O_3$ were measured using a UV photometer (Ansyco). NO and nitrogen dioxide ($NO_2$) concentrations were measured with a chemiluminescence analyzer (Eco Physics) equipped with a photolytic converter. Water vapor was monitored with a CRDS instrument (Picarro). Photolysis frequencies were calculated from actinic flux measurements by a spectroradiometer is placed on top of a building next to the

chamber. Calculations consider the transmittance of the chamber film and shading by the construction elements of the chamber (Bohn et al., 2005; Bohn and Zilken, 2005).

OH concentrations were measured by two instruments making use of either differential optical absorption spectroscopy (DOAS) (Dorn et al., 1995) or laser-induced fluorescence (LIF) (Fuchs et al., 2011). The DOAS instrument measured the absorption of light at 308 nm produced by a pico-second dye laser system. The 2 km long absorption path was folded in a

White cell along the long axis of the cylindrical shaped chamber.



The LIF instrument consists of three measurement cells for the separate detection of OH, HO$_2$, and RO$_2$ radicals. For the detection of OH, about 1100 cm$^3$ min$^{-1}$ of air is sampled into a low-pressure fluorescence cell, in which OH radicals are excited by a short laser pulse at 308 nm. The subsequent fluorescence signal is measured by a single photon-counting system (Fuchs et al., 2011).

HO$_2$ radicals were detected in a second LIF detection cell, after its conversion to OH in the reaction with NO. The concentration of NO is chosen, such that the formation of HO$_2$ from the concurrent conversion of RO$_2$ radicals is minimized (Fuchs et al., 2011). RO$_2$ concentrations are also indirectly measured in the ROxLIF system, where RO$_x$ (=RO$_2$+HO$_2$+OH) radicals in the sampled air are first converted to HO$_2$ by adding NO and CO in a conversion reactor. The air is then partly sampled into another low-pressure LIF detection cell, where HO$_2$ is converted to OH by excess NO followed by the measurement of OH by

fluorescence. The RO$_2$ concentration is finally derived from the difference in the radical measurements in the RO$_x$ and the HO$_x$ cells.

In three of the experiments, both the LIF and DOAS instruments were available. OH concentrations measured by the LIF instrument agreed with the concentrations measured by DOAS in two ozonolysis experiments (24 and 25 Janurary 2022, Table 2). The mean value of the difference between OH concentrations measured by the two instruments was about $0.7 \times 10^6$ cm$^{-3}$.

This is about the 1σ precision of measurements of the DOAS instrument (Table 1) and similar to the OH concentration during the experiments (Fig. 3 and Fig. S1). OH concentrations measured by the LIF instrument were used for the analysis of the ozonolysis experiments due to the higher time resolution and lower 1σ precision compared to that by the DOAS instrument.

OH concentrations measured by the LIF instrument were about 27 % higher than OH concentrations measured by the DOAS instrument in the photooxidation experiment (05 July 2022), when OH concentrations were well-above the 1σ precision of

measurements. Differences were larger than the combined 1σ accuracies of 15 % of the two instruments (7 % for the DOAS and 13 % for the LIF instruments). Because the DOAS instrument does not require calibration, its measurements are used for the analysis of the photooxidation experiments. The large difference might be due to an unaccounted calibration error of the LIF instrument.

OH radical concentrations were measured by only the DOAS instrument in the experiments on 06, 08 September 2021, 30

June 2022, and 05 July 2022. For the experiment on 06 July 2022, OH measurements were only available from the LIF instrument.

The OH reactivity in the chamber experiments was measured by a flash photolysis laser-induced fluorescence instrument as described in Section 3.1.

Sabinene was measured by proton-transfer-reaction mass spectrometry (Ionicon, PTR-TOF-MS). As the PTR-TOF-MS

instrument was not calibrated for sabinene, the initial sabinene concentration right after the injection was determined by the increase in the measured OH reactivity using the rate constant $k_{SAB+OH}$ (Section 4.1). The time series of the ion mass signal



measured by the PTR-TOF-MS instrument was then scaled to match this initial sabinene concentration to derive the time series of sabinene concentrations.

Acetone was also measured and calibrated by the PTR-TOF-MS instrument. HCHO concentrations were measured by a CRDS instrument (Picarro) and by the DOAS instrument that also detected OH radicals. HCHO concentrations measured by the DOAS and the CRDS agreed within 20 % in two of the experiments. In the experiment on 30 June 2022 a discrepancy of 45 % was observed. DOAS measurements were used for the analysis on that day as the CRDS methods requires correction factorss, whereas the DOAS method directly gibes concentration values (Glowania et al., 2021). On 6 July 2022, measurement by neither the DOAS nor the CRDS instrument was available.

Sabinaketone, which is one of the major products expected to be formed from the oxidation of sabinene, was detected as an uncalibrated ion-mass signal by the PTR-TOF-MS instrument (m/z: 139). Here, the sensitivity of nopinone, an isomer of sabinaketone, for which the PTR-TOF-MS instrument was calibrated, is used to determine the sabinaketone concentration assuming that the instrument has the same sensitivity for both compounds. This assumption has an uncertainty of 50 % based on Sekimoto et al. (2017).

**Table 1.** Instrumentation for radical and trace-gas measurements in the chamber experiments.

| Species | Method | Time resolution | 1-σ precision | 1-σ accuracy |
|---|---|---|---|---|
| OH | DOAS[a] | 205 s | $0.8 \times 10^6$ cm$^{-3}$ | 6.5 % |
| OH | LIF[b] | 47 s | $0.3 \times 10^6$ cm$^{-3}$ | 13 % |
| HO$_2$, RO$_2$ | LIF | 47 s | $1.5 \times 10^7$ cm$^{-3}$ | 16 % |
| OH reactivity | Laser flash photolysis + LIF | 180 s | 0.3 s$^{-1}$ | 0.5 s$^{-1}$ |
| NO | Chemiluminescence | 60 s | 20 pptv | 5 % |
| NO$_2$ | Chemiluminescence + photolytic converter | 60 s | 20 pptv | 5 % |
| O$_3$ | UV absorption | 180 s | 60 pptv | 5 % |
| Sabinene | PTR-TOF-MS[c] | 40 s | 10 % | 25 % |
| Sabinaketone | PTR-TOF-MS | 40 s | 10 % | 50 % |
| HCHO | DOAS | 100 s | 20 % | 7 % |
| HCHO | CRDS[d] | 300 s | 90 pptv | 10 % |
| Acetone | PTR-TOF-MS | 40 s | 5 % | 5 % |
| Photolysis frequencies | Spectroradiometer | 60 s | 10 % | 18 % |

[a] Differential optical absorption spectroscopy. [b] Laser-induced fluorescence. [c] Proton-transfer-reaction time-of-flight mass spectrometry. [d] Cavity ring-down spectroscopy.



### 3.4 Chamber experiments investigating the oxidation of sabinene

In total, seven experiments were performed to investigate the oxidation of sabinene by OH and $O_3$ under different conditions (Table 2). Two ozonolysis experiments were performed in winter 2022 at a low temperature of about 5 °C. In both experiments, the air was first humidified to a relative humidity of about 20 % (absolute humidity of 0.25 %). In the experiment on 24 January 2022 (Figure 3, Novelli et al., 2023a), 6 ppbv of sabinene was injected followed by the addition of 100 ppbv $O_3$ 30 minutes after the sabinene injection. Sabinene was oxidized for 3.5 hours by $O_3$ and partly by OH radicals that were produced from the ozonolysis reaction. Then, 100 ppmv of CO was injected as OH scavenger followed by another injection of 8 ppbv sabinene 30 minutes later, so that sabinene nearly exclusively reacted with $O_3$ for another 3.5 hours. The experiment on 25 January 2022 (Fig. S1, Novelli et al., 2023b) was performed in a similar way as the experiment on 24 January 2022 except that about 240 ppbv of $O_3$ was injected at the beginning of the experiment and the total number of sabinene injections was four instead of two.

Five photooxidation experiments were performed in summer 2021 and 2022 (Table 2). In each experiment, the chamber air was first humidified to a relative humidity between 50 % and 70 % (absolute humidity of 1.0 % to 2.0 %). Before the injection of sabinene, the chamber roof was opened to allow sunlight to irradiate the chamber air. No OH reactant was added during this part of the experiment (zero-air phase) to determine the production rate of chamber sources for HONO, NO, acetone, and HCHO.

Thirty minutes to two hours after opening the chamber roof, between 3 ppbv and 5 ppbv of sabinene was injected into the chamber and re-injected two to three times after most of the sabinene had been oxidized. In the experiments on 30 June 2022 (Fig. S2, Novelli et al., 2023c) and 06 July 2022 (Fig. S3, Novelli et al., 2023d), 60 ppbv and 120 ppbv of $O_3$ was injected into the chamber before opening the roof, respectively, to reduce the NO mixing ratios to less than 0.5 ppbv (denoted as 'low NO experiments'). In the experiments on 06 September 2021 (Fig. S4, Novelli et al., 2023e) and 05 July 2022 (Figure 4, Novelli et al., 2023f), there was no $O_3$ addition (denoted as 'medium NO experiments'), so that NO concentrations reached between 0.4 ppbv and 1.5 ppbv. The first part of the experiment on 08 September 2021 (Fig. S5, Novelli et al., 2023g) was like the other experiments with medium NO. In the second part of the experiment, however, 60 ppbv of $O_3$ was injected, such that NO mixing ratios were suppressed and sabinene reacted under similar conditions like in the experiments with low NO mixing ratios. In the experiments with medium NO mixing ratios, sabinene almost exclusively (> 90 %) reacted with OH. In the experiments with low NO mixing ratios, only about 60 % to 80 % of sabinene reacted with OH and the remaining part reacted with $O_3$.



**Figure 3.** Overview plot of measured radical and trace gas concentrations in the ozonolysis experiment performed on 24 January 2022 (Novelli et al., 2023a). PTR-TOF-MS measurements of sabinene were derived from scaling the ion mass signal to the increase of the OH reactivity right after the injections. The contributions of different pathways to the total loss rate constant of RO$_2$ radicals, $k_{RO2}$, are calculated from the reactivity of RO$_2$ radicals to bimolecular reactions using measured trace gas concentrations (Supplementary material Section 1). The black vertical dotted line indicates the time when sabinene was injected and the red vertical dotted line indicates the time when CO was injected. After the injection of CO, the OH reactivity was too high to be measured (~ 500 s$^{-1}$).



**Figure 4.** Overview plot of measured radical and trace gas concentrations in a photooxidation experiment with medium NO mixing ratios performed on 05 July 2022 (Novelli et al., 2023f). PTR-TOF-MS measurements of sabinene were derived from scaling the ion mass signal to the increase of the OH reactivity right after the injection. The contributions of different pathways to the total loss rate constant of $RO_2$ radicals, $k_{RO2}$, are calculated from the reactivity of $RO_2$ radicals to bimolecular reactions using the measured trace gas concentrations (Supplementary material Section 1). Vertical dotted lines indicate times when sabinene was injected.





**Table 2.** Summary of conditions of experiments performed in this study.. For temperature NO, OH, and $O_3$ concentrations, they are given as the range of mean values when sabinene was present in the chamber. The range of sabinene mixing ratios represents the range of maximum values reached right after each injection.

| Type of experiment | Temperature (K) | NO (ppbv) | OH ($10^6$ cm$^{-3}$) | $O_3$ (ppbv) | Sabinene (ppbv) | Date | Figure | Reference |
|---|---|---|---|---|---|---|---|---|
| Ozonolysis | 278 – 280 | 0 | < 1 | 105 | 6 | 24 January 2022 | Figure 3 | Novelli et al. (2023a) |
| | 276 – 277 | 0 | < 1 | 220 | 3 – 6 | 25 January 2022 | Fig. S1 | Novelli et al. (2023b) |
| Low NO | 300 – 305 | 0.15 – 0.2 | 5 – 7 | 70 – 80 | 4 – 4.5 | 08 September 2021 (2nd and 3rd injections) | Fig. S5 | Novelli et al. (2023g) |
| | 299 – 307 | 0.2 – 0.3 | 3 – 4 | 60 – 75 | 4 – 8 | 30 June 2022 | Fig. S2 | Novelli et al. (2023c) |
| | 293 – 297 | 0.05 – 0.15 | 2 – 5 | 100 – 110 | 4 – 6 | 06 July 2022 | Fig. S3 | Novelli et al. (2023d) |
| Medium NO | 300 – 303 | 0.4 – 0.6 | 4 – 6 | 5 – 25 | 3.5 | 06 September 2021 | Fig. S4 | Novelli et al. (2023e) |
| | 294 – 299 | 0.5 | 3 – 6 | 0 – 15 | 4 | 08 September 2021 (1st injection) | Fig. S5 | Novelli et al. (2023g) |
| | 303 – 305 | 0.5 – 1.5 | 2 – 5 | 10 – 40 | 4 – 6 | 05 July 2022 | Figure 4 | Novelli et al. (2023f) |

## 3.5 Calculation of yields of oxidation products

Product yields from the oxidation of sabinene are calculated by performing a linear regression between the concentrations of product species and the amount of sabinene that was oxidized. The measured time series of product concentrations for acetone, HCHO, and sabinaketone were corrected for their additional production from the chamber and their loss due to dilution, photolysis and reaction with OH radicals as described in Kaminski et al. (2017) and Rolletter et al. (2019). Photolysis rates and dilution rates were measured and reaction rate constants with OH radicals for acetone and HCHO were retrieved from IUPAC recommendations (Atkinson et al., 2006). The reaction rate constant of the reaction of sabinaketone and OH radicals is taken from measurements by Alvarado et al. (1998) and Carrasco et al. (2007) giving a value of $6 \times 10^{-12}$ cm$^3$ s$^{-1}$. The potential loss of sabinaketone by photolysis is estimated using photolysis rates of ketones calculated in the MCM model. The maximum photolysis rate loss constant for ketones during the experiments was $3 \times 10^{-6}$ s$^{-1}$, which would result in only a negligible loss of sabinaketone yield.





**3.6 Calculation of the reaction rate constants and OH yield in the ozonolysis of sabinene in the chamber experiments**

The rate constant of the ozonolysis reaction of sabinene ($k_{SAB+O3}$), and the OH yield ($\gamma_{SAB}$) of this reaction were determined from the ozonolysis experiment. Sabinene was lost by its reactions with OH and $O_3$, and by dilution when no OH scavenger was present. The loss rate is described by the differential equation:

$$\frac{d[SAB]}{dt} = -[SAB](k_{SAB+O3}[O_3] + k_{SAB+OH}[OH] + k_{dil})$$ (3)

In the presence of an OH scavenger, the sabinene loss in the reaction with OH becomes zero and the reaction rate constant

$k_{SAB+O3}$ can be directly determined from a fit of measured time series of the sabinene concentration using Eq. (3). Equation (3) can be simplified by replacing the time-dependent quantities by their mean values over the time interval when sabinene was oxidized:

$$\frac{d[SAB]}{dt} = -[SAB](t)(k_{SAB+O3}\langle[O_3]\rangle_t + k_{SAB+OH}\langle[OH]\rangle_t + \langle k_{dil}\rangle_t)$$ (4)

In this approximation, the temporal evolution of the sabinene concentrations simplifies to a single-exponential decay with a loss rate constant $k_{loss}$:

$$[SAB](t) = [SAB]_0 \exp(-k_{loss}t) = [SAB]_0 \exp\left(-(k_{SAB+O3}\langle[O_3]\rangle_t + k_{SAB+OH}\langle[OH]\rangle_t + \langle k_{dil}\rangle_t)t\right)$$ (5)

The value of the loss rate constant $k_{loss}$ can be experimentally determined from a fit to the measured sabinene concentrations.

The use of time-averaged values can be justified as the $O_3$ concentrations during the ozonolysis experiments decreased by only about 10 % between consecutive sabinene injections mainly due to dilution. Therefore, the additional systematic error in the calculation of the total loss rate constant of sabinene from using the mean $O_3$ concentrations is only about 5 %. The error from using the time-averaged dilution rate constant is insignificant as less than 5 % of sabinene was lost by dilution. To determine

the OH yield of the sabinene ozonolysis reaction $\gamma_{SAB}$, the OH concentration in Eq. (5) is expressed as a function of the OH production rate ($P_{OH}$) and its loss rate. The loss rate of OH can be expressed as the product of OH reactivity $k_{OH}$ and the OH concentration. Due to the short lifetime of the OH radical, the production rate of OH is always balanced by the destruction rate of OH. Here, it is assumed that OH radicals are only produced from the ozonolysis of sabinene and the reaction of $O_3$ with $HO_2$ radicals. The OH reactivity can be assumed to be only from sabinene right after its injection as contributions from species

other than sabinene ($O_3$, sabinaketone, HCHO, acetone etc.) to the total OH reactivity were less than 10 % until half of the injected sabinene had reacted away. Therefore, only this time interval, which is about the first hour after the injection of sabinene, is used for this evaluation. Overall, the OH concentration can be expressed as

$$[OH] = \frac{P_{OH}}{k_{OH}} \approx \frac{k_{HO2+O3}[O_3][HO_2] + k_{SAB+O3}[O_3][SAB]\gamma_{SAB}}{k_{SAB+OH}[SAB]}$$ (6)



By combining Eq. (5) and (6), the total loss rate constant of sabinene to chemical reactions ($k_{\text{loss,chem}}$) in the ozonolysis experiment without an OH scavenger can be expressed as a function of the reaction rate constant $k_{\text{SAB+O3}}$ and the OH yield

$\gamma_{\text{SAB}}$, after correcting for the additional OH production from the reaction between $HO_2$ and $O_3$.

$$k_{\text{loss,chem}} \text{ (without CO)} = \frac{k_{\text{loss}} - \langle k_{\text{dil}} \rangle_t - \frac{k_{\text{HO2+O3}} \langle [O_3] \rangle_t \langle [HO_2] \rangle_t}{\langle [SAB] \rangle_t}}{\langle [O_3] \rangle_t} = k_{\text{SAB+O3}}(1 + \gamma_{\text{SAB}}) \tag{7}$$

When the OH scavenger is present, the loss rate constant of sabinene to chemical reactions is the reaction rate constant of the ozonolysis reaction:

$$k_{\text{loss,chem}}(\text{with CO}) = \frac{k_{\text{loss}} - \langle k_{\text{dil}} \rangle_t}{\langle [O_3] \rangle_t} = k_{\text{SAB+O3}} \tag{8}$$

The reaction rate constant $k_{\text{SAB+O3}}$ was first determined in the experiments with OH scavenger using Eq. (8) and was then used to determine the OH yield in the experiments when the OH scavenger was not present using Eq. (7).

The uncertainty of the reaction rate constant $k_{\text{SAB+O3}}$ stems from the uncertainty in the fitted loss rate $k_{\text{loss}}$ (5 % to 9 %) and from the uncertainty of the $O_3$ measurement (5 %, Table 1). Therefore, the reaction rate constant $k_{\text{SAB+O3}}$ has a total uncertainty of about 11 %. The uncertainty of the OH yield $\gamma_{\text{SAB}}$, which depends on the uncertainty of the loss rate constants $k_{\text{loss,chem}}$, can be as high as 50 % (Table S1). This is because the loss of sabinene by OH radicals is less important than the loss of sabinene by the ozonolysis reaction and the uncertainty of the OH yield is enhanced by the uncertainty of the ozonolysis reaction rate

constant $k_{\text{SAB+O3}}$. It is worth noting that potential systematic errors of about -20 % is not considered in the calculation, which arises from the assumption that the entire OH reactivity $k_{\text{OH}}$ results from only sabinene (Eq. (6)), and from the temperature difference of 2 °C to 4 °C between the ozonolysis experiments with and without an OH scavenger. The value of the ozonolysis reaction rate constant $k_{\text{SAB+O3}}$ could be lower by 10 to 15 % at a temperature at 276 K than at temperature of 280 K. This is estimated from the temperature dependence of ozonolysis rate constants of structurally similar methylpropene and β-pinene,

for which ozonolysis temperature-dependent reaction rate constants are reported in Cox et al. (2020).

The rate constant of the reaction of sabinene with OH $k_{\text{SAB+OH}}$ was determined using measurements from the experiments with medium NO mixing ratios (Table 2), because the contribution of the OH reaction to the total loss of sabinene was more than 90 % in these experiments. The rate constant $k_{\text{SAB+OH}}$ was determined by minimizing the root-mean-square error between sabinene concentrations measured by the PTR-TOF-MS instrument and calculations using a simplified chemical model as

described in Hantschke et al. (2021). The simplified model includes the chemical loss of sabinene by the reactions with OH and $O_3$ and by dilution. OH and $O_3$ concentrations were constrained to measurements. No further other secondary chemistry was included. The rate constant of the ozonolysis reaction in the simplified model was taken from the recommended value ($8.3 \times 10^{-17}$ cm$^3$ s$^{-1}$, Cox et al., 2020), as the temperature during the experiments in this study with medium NO mixing ratios was similar to that of the experiments reported in the literature, whereas the temperature in the ozonolysis experiment in this

work was 20 °C lower. In the simulation, the OH concentration was constrained to measurements by either the DOAS or LIF



instrument, depending on the availability of instruments. A value of the reaction rate constant $k_{SAB+OH}$ was obtained for each injection of sabinene in the experiments with medium NO mixing ratio and each available OH instrument. The mean and standard deviation of the rate constant are then calculated with values from every injection obtained from an instrument.

## 3.7 Analysis of the chemical budget of OH radicals

Measurements in the chamber experiments were also used to study the chemical budget of OH radicals in the ozonolysis experiment and the photooxidation experiments at different NO mixing ratios. Since the lifetimes of OH is very short (< 1 s), production rate of OH radical must equal its destruction rate on the time scale of the experiment. By considering radical production and destruction pathways that are typically included in atmospheric chemistry models, insights can be provided into if there are missing radical production pathways for example from fast $RO_2$-isomerization reactions (e.g., Fuchs et al., 2013; Novelli et al., 2020), or if the rates of radical destruction pathways are underestimated (Pang et al., 2022). Table 3 lists all reactions producing OH radicals that were considered in the chemical budget analysis in this work. The destruction rate of OH radicals was calculated from the product of OH reactivity and OH concentration measurements (Hofzumahaus et al., 2009).

**Table 3.** Reactions producing OH radicals considered in the analysis of the chemical budget of radicals. Unless specified, reaction rate constants are given for room temperature (T = 298 K) and 1 atm pressure.

| Reaction | $k$ | 1σ uncertainty[a] (%) | Reference |
|---|---|---|---|
| HONO + $h\nu$ → OH + NO | $j_{HONO}$ | 35[b] | Measured |
| $O_3$ + $h\nu$ + $H_2O$ → 2 OH + $O_2$ | $\phi_{OH}$[c], $j_{O3}$ | 19 | Measured |
| $HO_2$ + NO → OH + $NO_2$ | $8.8 \times 10^{-12}$ cm$^3$ s$^{-1}$ | 20 | Atkinson et al. (2004) |
| $HO_2$ + $O_3$ → OH + $2O_2$ | $2.0 \times 10^{-15}$ cm$^3$ s$^{-1}$ | 28 | Atkinson et al. (2004) |
| sabinene + $O_3$ → 0.18OH + 0.18$RO_2$ | $8.3 \times 10^{-17}$ cm$^3$ s$^{-1}$ (298 K) | 16 | Cox et al. (2020) |
| | $3.7 \times 10^{-17}$ cm$^3$ s$^{-1}$ (278 K) | 11 | This study |

[a]Total 1σ uncertainty of the reaction: including uncertainties from measurements, reaction rate constants and OH yield from ozonolysis. [b]HONO was not measured in all experiments, but its concentrations were calculated from OH, NO, and $j_{HONO}$ measurements during the zero-air phase, the uncertainty in the HONO concentration is about 30 %. [c]yield of OH radicals of the photolysis of $O_3$.





## 4. Results and Discussions

### 4.1 Rate constants of the reactions of sabinene with OH and O₃

The average value of the ozonolysis reaction rate constant $k_{SAB+O3}$ determined from the chamber experiments in this work is $(3.7\pm0.5)\times10^{-17}$ cm$^3$ s$^{-1}$ (Table S1). This value is 55 % lower than values reported in the literature (Table 4), in which the

reaction rate constant $k_{SAB+O3}$ was determined at room temperature $((296\pm2)$ K) using absolute and relative rate techniques (Atkinson et al., 1990a, b; Bernard et al., 2012). The lower value determined in this study could be due to the low temperature (278 K) in the chamber experiments. This is supported by the know temperature dependence of ozonolysis rate constants of structurally similar alkenes such as isobutene, β-pinene, and camphene, for which values decrease by about 25 % to 50 % with relative uncertainties of about 25 % (Cox et al., 2020).

SAR in Jenkin et al. (2020) gives four to five times lower values for the ozonolysis reaction rate constant of sabinene than values determined in this work and reported in literature. Values are $1.4\times10^{-17}$ and $9.3\times10^{-18}$ cm$^3$ s$^{-1}$ at 298 K and 278 K, respectively. Since all experimentally determined values are higher, it is likely that SAR underpredicts the rate constant $k_{SAB+O3}$.

The Arrhenius expression of the OH reaction rate constant $k_{SAB+OH}$ derived from OH reactivity measurements at temperature between 284 K and 340 K at ambient pressure in this work (Section 3.1) is

$k_{SAB+OH}(T) = (1.67\pm0.16)\times10^{-11}\times\exp((537\pm30)\ T^{-1})$ cm$^3$ s$^{-1}$

The accuracy is 13 % to 15 % (Figure 5), which is mainly due to the uncertainty of sabinene concentrations during the OH reactivity measurement. The value at room temperature (T = 298 K) is $(1.0\pm0.2)\times10^{-10}$ cm$^3$ s$^{-1}$. This agrees with the value required to describe the consumption of sabinene in the chamber experiments of $(1.4\pm0.5)\times10^{-10}$ cm$^3$ s$^{-1}$ (Fig. S6), as well as the value of $(1.17\pm0.05)\times10^{-10}$ cm$^3$ s$^{-1}$ determined in laboratory experiments Atkinson et al. (1990a). The temperature-

dependence coefficient of the rate constant $k_{SAB+OH}$ of $(537\pm30)$ K is similar to that of structurally-similar β-pinene $((460\pm150)$ K) and isobutene $((505\pm200)$ K) (Mellouki et al., 2021).





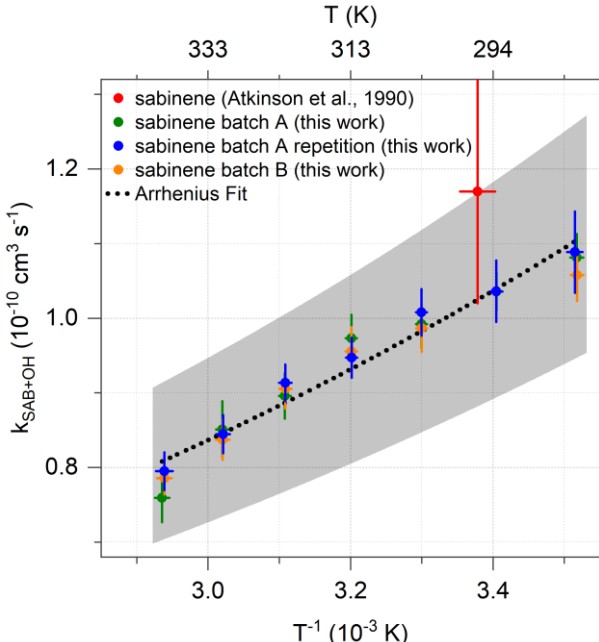

**Figure 5.** The OH reaction rate constant $k_{SAB+OH}$ determined in laboratory experiments using the OH reactivity measurements. The shaded area represents the accuracy of the Arrhenius expression.


**Table 4.** List of rate constants of the reaction of sabinene with OH and $O_3$ reported in literature and determined in this study.

|  | Reaction rate constant ($cm^3$ $s^{-1}$) | Temperature (K) | Method | Reference |
|---|---|---|---|---|
| sabinene + OH | $(1.17\pm0.05) \times 10^{-10}$ | $296\pm2$ | Relative rate | Atkinson et al. (1990a) |
|  | $6.08\times10^{-11}$ | 298 | SAR | Jenkin et al. (2018) |
|  | $(1.67\pm0.16)\times10^{-11}\times\exp((537\pm30)T^{-1})$ | 284 – 340 | OH reactivity measurements | This study |
| sabinene + $O_3$ | $(8.1\pm0.8) \times 10^{-17}$ | $296\pm2$ | Absolute rate | Atkinson et al. (1990a) |
|  | $(6.2\pm2.1) \times 10^{-17}$ | $297\pm2$ | Absolute rate | Bernard et al. (2012) |
|  | $(8.8\pm1.0) \times 10^{-17}$ | $296\pm2$ | Relative rate | Atkinson et al. (1990b) |
|  | $1.4\times10^{-17}$ | 298 | SAR | Jenkin et al. (2020) |
|  | $(3.7\pm0.5) \times 10^{-17}$ | $278\pm2$ | Absolute rate | This work |



**4.2 Product yields of organic compounds from the oxidation of sabinene**

Formaldehyde is one of the major products from the oxidation of sabinene by both oxidants, OH and $O_3$. The analysis of the photooxidation experiments results in a HCHO yield of (46±25) %. The uncertainty is due to the uncertainty of the HCHO chamber source, corrections for the loss of HCHO, and the calculation of the amount of reacted sabinene. There is no significant difference in the HCHO yields determined in the experiments with either medium or low NO mixing ratios (Fig. 6a), which can be expected as the majority of $RO_2$ radicals reacted with NO in both experiments (Fig. S2 and Figure 4). The HCHO yield

reported in this study is higher than the yield of 25 % reported by Carrasco et al. (2006a) (Table 5), who conducted experiments at a high concentration (~ 2 ppmv) of sabinene using either $H_2O_2$ or HONO photolysis as OH sources. The authors found similar HCHO yields in their experiments for different NO concentrations. Larsen et al. (2001) also performed experiments in the absence of NO and they a HCHO yield of (35±4) %. This value is lower than the yield found in this study but still agrees within the uncertainties.

The photooxidation of sabinene results in an acetone yield of (21±15) % in this study (Figure 6b), when OH radicals predominantly reacted with sabinene. The uncertainty of the acetone yield is mainly due to the uncertainty in the chamber source, corrections for losses, and the calculation of the amount of reacted sabinene. The value is consistent with the yields in the studies by Carrasco et al. (2006) and Reissell et al. (1999) (Table 5). Reissell et al. (1999) performed photooxidation experiments at high initial NO concentrations (~10 ppmv) and with a high NO:sabinene concentration ratio (~10:1). In another

study by Larsen et al. (2001), the acetone yield was lower than the values found in this study and studies conducted by Reissell et al. (1999) and Carrasco et al. (2006).

The yield of sabinaketone of the photooxidation of sabinene is (19±16) % (Figure 6c). The large uncertainty of the calculation is due to the uncertainty of the measurement sensitivity of sabinaketone. This value agrees well with literature values that range from 17 % to 24 % (Table 5). There is no significant dependence of the yield of sabinaketone from the OH oxidation of

sabinene on the NO mixing ratio, which is also consistent with the findings in Carrasco et al. (2006).

The HCHO yield determined in the ozonolysis experiments is (48±15) %, when the OH oxidation was suppressed by the presence of an OH scavenger. This value is consistent with the yield of (52±9) % reported by Chiappini et al. (2006), who conducted ozonolysis experiments in the presence of an OH scavenger and with high concentrations (~ 1 ppmv) of sabinene and $O_3$.

The ozonolysis of sabinene produces a small amount of acetone compared to the photooxidation by OH. The low acetone yield of (5±2) % determined in this study agrees well with those reported by Reissell et al. (1999) and Chiappini et al. (2006) (Table 5).

The yield of sabinaketone from the ozonolysis of sabinene is (31±15) %. This value is lower than values reported in literature, which range between 35 % to 50 % , but still agrees within the combined uncertainties (Hakola et al., 1994; Yu et al., 1999;

Chiappini et al., 2006, Table 5). The sabinaketone yield could increase with increasing humidity due to the reaction of the

 

stabilized Criegee intermediates with water (Wang and Wang (2017), Fig. 2) and a laboratory study on the nopinone yield from structurally-similar β-pinene ozonolysis (Ma and Marston, 2008). However, the absolute humidity during the ozonolysis experiments in this study of 0.25 % was not much different or higher than the humidity in the experiments reported in literature (Table 5), so that it is unlikely that humidity explains the lower value in this work.


**4.3 Production of OH radicals from sabinene ozonolysis**

The OH yield from the sabinene ozonolysis reaction can be calculated by comparing the total loss rate constants of sabinene in the presence and absence of an OH scavenger (Eq. (**7**) and (**8**), Table S1). An OH yield of (18±25) % is obtained from the experiments in this work. OH is not only produced from the ozonolysis reaction, but about 30 to 40% is produced from the reaction of $HO_2$ with $O_3$ within the timeframe of the analyzed experiment. Productions of $HO_2$ radicals were observed once sabinene started reacting with $O_3$, which was likely relate to the reaction between OH and $O_3$, as well as the reactions of $RO_2$ with other $RO_2$ radicals . The uncertainty of the OH yield is large, because the additional loss of sabinene from the reaction with OH is small compared to the loss of sabinene from the ozonolysis reaction. The uncertainty of the ozonolysis rate constant $k_{SAB+O3}$ further amplifies the uncertainty of the OH yield $\gamma_{SAB}$ (Table S1).

The value determined in this work agrees with those reported in the literature of (26±13) % (Atkinson et al., 1992) and (33±5) % (Aschmann et al., 2002). Calculating the OH yield $\gamma_{SAB}$ without accounting for the contribution from $HO_2+O_3$, as in previous studies due to the lack of $HO_2$ measurements, would result in a value of (30±22) % for experiments in this work (Table S1). This value is still in agreement with both literature values (Table 5).

**Table 5.** Summary of the product yields from the oxidation of sabinene reported in literature and determined in experiments in this study.

| | Acetone | HCHO | Sabinaketone | OH | Reference |
|---|---|---|---|---|---|
| Sabinene + OH | / | / | 17 % | NA | Arey et al. (1990) |
| | / | / | (17±3) % | NA | Hakola et al. (1994) |
| | (19±3) % | / | / | NA | Reissell et al. (1999) |
| | (9±3) % | (35±4) % | (24±10) % | NA | Larsen et al. (2001) |
| | (25±5) % | (25±5) % | (22±6) % (no NOx) | NA | Carrasco et al. (2006) |
| | (23±5) % | (25±6) % | (19±5) % (with NOx) | NA | Carrasco et al. (2006) |
| | 45 % | 47 % | 47 % | NA | Wang and Wang (2018)[f,g] |
| | (21±15) % | (46±25) % | (19±16) % | NA | This work |



| Sabinene + $O_3$ | $(3\pm2)$ % | / | / | / | Reissell et al. (1999) |
|---|---|---|---|---|---|
| | / | / | $(47\pm24)$ %[a] | / | Yu et al. (1999) |
| | / | / | $(50\pm9)$ %[b] | / | Hakola et al. (1994) |
| | Detected | $(52\pm9)$ % | $(35\pm14)$ %[c] | / | Chiappini et al. (2006) |
| | / | / | / | $(33\pm5)$ % | Aschmann et al. (2002) |
| | / | / | / | $(26\pm13)$ % | Atkinson et al. (1992) |
| | / | 83 % | 17 % (dry) | 44 % | Wang and Wang (2017)[f] |
| | / | 83 % | 47 % (humid)[d] | 44 % | Wang and Wang (2017)[f] |
| | $(5\pm2)$ % | $(48\pm15)$ % | $(31\pm15)$ %[e] | $(18\pm25)$ % | This work |

NA: Not applicable [a]Experiments were performed at around 5 % relative humidity (Griffin et al., 1999) [b]The humidity is not mentioned [c]Experiments were performed with less than 300 ppm of water [d]Calculated for 298 K and 50 % relative humidity. [e]Ozonolysis experiments were conducted at 0.25 % absolute humidity. [f]Theoretical calculations [g]Assuming high $NO_x$ concentrations and $RO_2$ + NO is the dominating loss for $RO_2$ radicals.






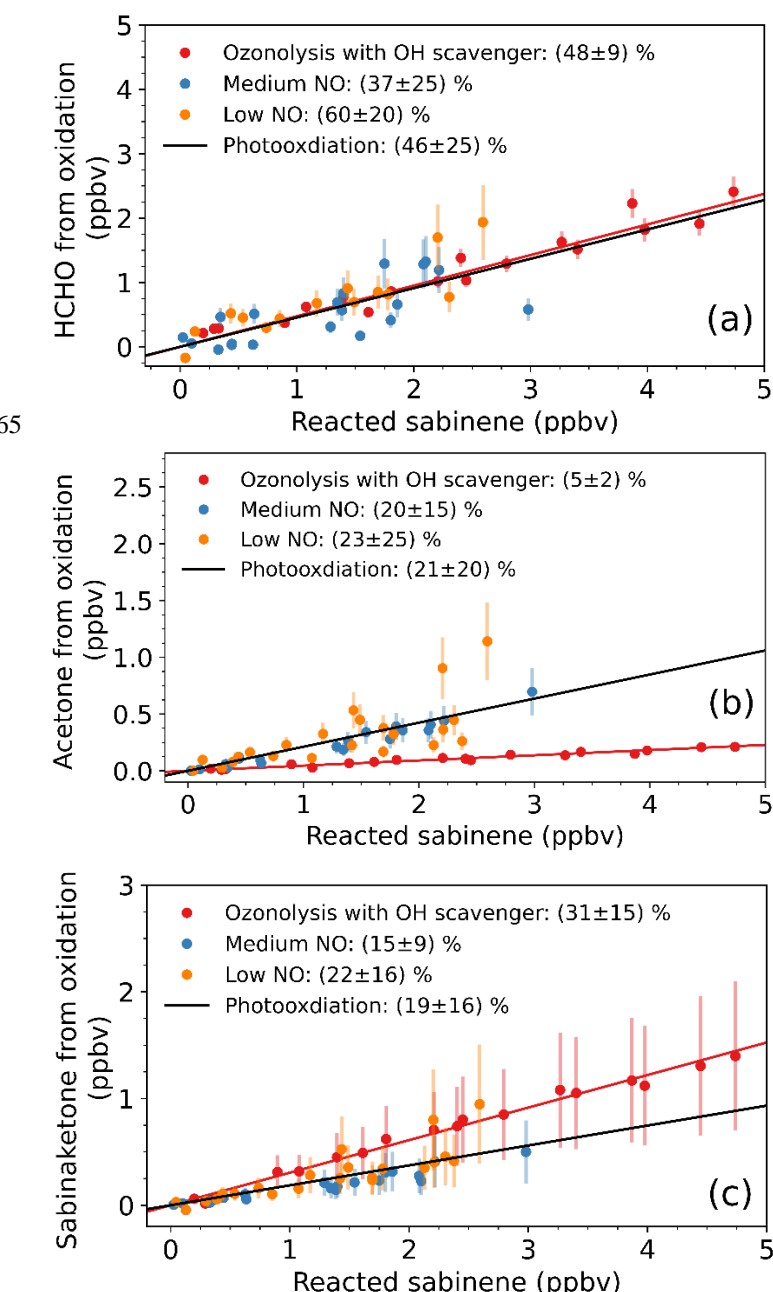


**Figure 6.** Determination of the yields of HCHO (a), acetone (b), and sabinaketone (c) from the reaction of sabinene with OH and O₃. 'Photooxidation' refers to the yields calculated from all data in the experiments with low and medium NO mixing

ratios. For clarity, only data points from the time period, when 50 % of the injected sabinene was still present (3 ppbv to 4 ppbv) are shown. Fig. S7 shows the plots with the analysis using all data points.





## 4.4 Comparison of experimental results with values expected from the theoretically determined sabinene oxidation mechanism

The product yields of the photooxidation of sabinene determined in this study can be compared to yields expected from the sabinene oxidation mechanisms by Wang and Wang (2017) and Wang and Wang (2018).

In the OH oxidation mechanism by Wang and Wang (2018), HCHO is only produced from the subsequent chemistry of the RO$_2$ radical SABINOHAO2 that results from one of the two OH-addition reactions of sabinene (Figure 1). The HCHO yield of (46±25) % (Table 5) determined in the photooxidation experiments is consistent with the branching ratio of 47 % of the

respective OH-addition reaction channel in the mechanism (reaction pathway (a) in Figure 1).

The yield of acetone in the OH oxidation of sabinene expected from the mechanism by Wang and Wang (2018) is determined by the branching ratio of the OH-addition reaction producing the RO$_2$ radical SABINOHBO2 as well as by the fraction of SABINOHBO2 that undergoes an isomerization reaction, from which eventually acetone is produced. For atmospheric conditions like in the experiments in this work, it is expected that more than 90 % of SABINOHBO2 undergoes the

unimolecular reaction ($k_{uni}$ ~ 5 s$^{-1}$, Figure 1) and less than 10 % undergoes bimolecular reactions ($k_{RO2}$ < 0.4 s$^{-1}$, Figure 4 and Figs. S2 and S3). Therefore, the acetone yield is expected to be similar as the branching ratio for the second OH addition reaction of 45 % (reaction path (b) in Figure 1).

However, the experimentally determined acetone yield of (21±15) % is significantly lower than this value. To bring both values into agreement, the rate constant of the unimolecular reaction of SABINOHBO2 would need to be in the order of magnitude

of the loss rates constant of bi-molecular RO$_2$ reactions in this study (~0.2 s$^{-1}$), such that about half of the RO$_2$ radical SABINOHBO2 undergoes bi-molecular reactions (mainly with NO). Another reason for the discrepancy could also be that products other than acetone are partly formed from the uni-molecular reaction pathway of RO$_2$ radical SABINOHBO2. This would be consistent with the observation that the acetone yield found in this study and in Carrasco et al. (2006) does not strongly depend on the NO concentration (Table 5), which would be expected, if the bi-molecular RO$_2$ reactions with for

example NO were competitive with the isomerization reaction.

The sabinaketone yield of (19±16) % from the OH-oxidation of sabinene determined from the experiment is lower than yield of 47 % expected from the mechanism by Wang and Wang (2018). The mechanism by Wang and Wang (2018) also suggests the production of sabinaketone comes together with the production of HCHO and thereby have the same yield. However, only the HCHO yield determined in this study agrees with the HCHO yield expected from the mechanism by Wang and Wang

(2018). The large difference between the HCHO yield and sabinaketone yield in this study could be due to the large uncertainties of the yields.

Regarding the ozonolysis of sabinene, the HCHO yield determined in the experiments in this study and in the study by Chiappini et al. (2006) are about 35 % lower than the HCHO yield of 83 % expected from the mechanism by Wang and Wang (2017). In their mechanism, HCHO is directly formed as co-product of the two Criegee intermediates CI-1 and CI-2, so that



the HCHO yield reflects the sum of the branching ratios for these reaction pathways (reactions (B) and (C) in Figure 2). The low HCHO yield in the experiments might therefore hint that the branching ratios in the mechanism by Wang and Wang (2017) are too high.

The production of acetone from the ozonolysis of sabinene is not discussed in Wang and Wang (2017). The small yield of $(5\pm2)$ % determined in this work also suggest that only minor reactions pathways lead to the production of acetone. One

feasible mechanism could be the breakage of the 3-membered ring in the Criegee intermediate CI-2 that yields a biradical BI-RAD (Figure 7). A similar mechanism was proposed for the Criegee intermediates from the ozonolysis of β-pinene (Nguyen et al., 2009). The expected yield of the analogous biradical formed from the ozonolysis of β-pinene is 3 %, which explains the low acetone yield between 1 % to 7 % observed for β-pinene (Lee et al., 2006). If a similar reaction pathway applies for Criegee intermediate from sabinene, it can be expected that this is mainly relevant for the Criegee intermediate CI-2 because of the fast

loss of the Criegee intermediate CI-1 due to the fast H-migration reaction (Wang and Wang, 2017). Therefore, the small acetone yield might be explained by the breakage of the 3-membered ring of the Criegee intermediate CI-2.

The sabinaketone yield of $(31\pm15)$ % determined from the ozonolysis experiments is lower than the value of 47 % expected from the calculations in Wang and Wang (2017). As discussed above in the comparison with values reported in literature, humidity could impact the sabinaketone yield, but the overall effect is expected to be small. Values reported in literature are

in better agreement with the yield of 47% expected from the mechanism by Wang and Wang (2017). Therefore, the low value determined in the work is likely due to the high uncertainty in the sabinaketone measurements.

The OH yield from the ozonolysis reaction of sabinene of $(18\pm25)$ % determined in the experiments is lower than the yield of 44 % expected from the mechanism in Wang and Wang (2017). In the mechanism, OH is mainly produced from the unimolecular reaction of the Criegee intermediate CI-1 (Figure 2). The large difference between the OH yield determined in

the experiments and expected from the mechanism could indicate that the branching ratio of pathway (C) producing CI-1 is too high (Figure 2).

The yield of the stabilized Criegee intermediate sCI-1 can also affect the expected OH yield in the mechanism. Calculations by Wang and Wang (2017) suggest that there is a high uncertainty of the yield of the stabilized Criegee intermediate sCI-1. Laboratory experiments quantifying the yield of the stabilized Criegee intermediate would partly help constraining the OH

yield. However, from the best of our knowledge there are no measurements of stabilized Criegee intermediates from the ozonolysis of sabinene.



**Figure 7.** Possible reaction pathway of the Criegee intermediate CI-2 that could explain the small production of acetone from the ozonolysis of sabinene.

## 4.5 Chemical budget of OH radicals

The sum of OH radical production rates from different pathways needs to be balanced by the OH destruction rate. In the ozonolysis experiment (Figure 8a), the production and destruction rates of OH were low with values of less than 1.5 ppbv h$^{-1}$. The OH concentration was close to the limit of detection of the instrument ($< 10^6$ cm$^{-3}$), so that the calculations have a high uncertainty. In the first hour of the experiment, about 65 % of the total OH production was from the ozonolysis reaction (0.75 ppbv h$^{-1}$) and the remaining part was from the reaction of HO$_2$ with O$_3$ (0.5 ppbv h$^{-1}$). The total production rate of OH was on average slightly higher (+0.2 ppbv h$^{-1}$) than the destruction rate. However, this discrepancy is within the uncertainty of the calculations, so that the chemical budget can be regarded as balanced by the considered OH production and destruction reactions.

In the photooxidation experiment with low NO mixing ratios (Figure 8b), the production rate of OH ranged between 2 and 6 ppbv h$^{-1}$. A good agreement between the OH production and destruction rates can be seen during the zero-phase demonstrating that the analysis includes all important OH production pathways in the clean chamber before sabinene was injected.

After the injection of sabinene, the OH destruction rate increased due to the consumption by sabinene. The OH production rate concurrently increased due to the enhanced regeneration of OH from the reaction of HO$_2$ with NO and due to the production of OH from the ozonolysis of sabinene. For NO mixing ratios (0.05 ppbv to 0.15 ppbv) in this experiment, 20 % to 60 % of the OH was produced from OH regeneration in the reaction of HO$_2$ with NO, and the remaining part was from the photolysis of HONO and O$_3$ and the reaction of HO$_2$ with O$_3$. The contribution of the ozonolysis of sabinene to the total OH production was maximum 10 % to 30 % right after both sabinene injections, but quickly decreased, while sabinene was being oxidized. Overall, the OH production rate was excellently balanced by the OH destruction rate in the experiments with low NO mixing ratios suggesting that there was no significant missing OH sources for the conditions of these experiments.

In the photooxidation experiment with medium NO concentrations (Figure 8c), the OH production and destruction rates were 4 ppbv to 10 ppbv h$^{-1}$ which was higher than values in the experiments with low NO. The high production rate is mainly due to a fast regeneration of OH in the reaction of HO$_2$ with NO contributing 70 % to 80 % to the total OH production rate. The production rate of OH was also well balanced by the OH destruction rate within the uncertainty of the calculations in this experiment.

In summary, the OH production and destruction rates were well-balanced for all conditions experienced in this work. This suggests that there are no unaccounted OH production reactions in the photooxidation of sabinene, so that well-known



photolysis reactions and bi-molecular reactions are sufficient to be considered in the chemical budget of OH radicals. This is

consistent with the mechanism proposed by Wang and Wang (2017 and 2018).

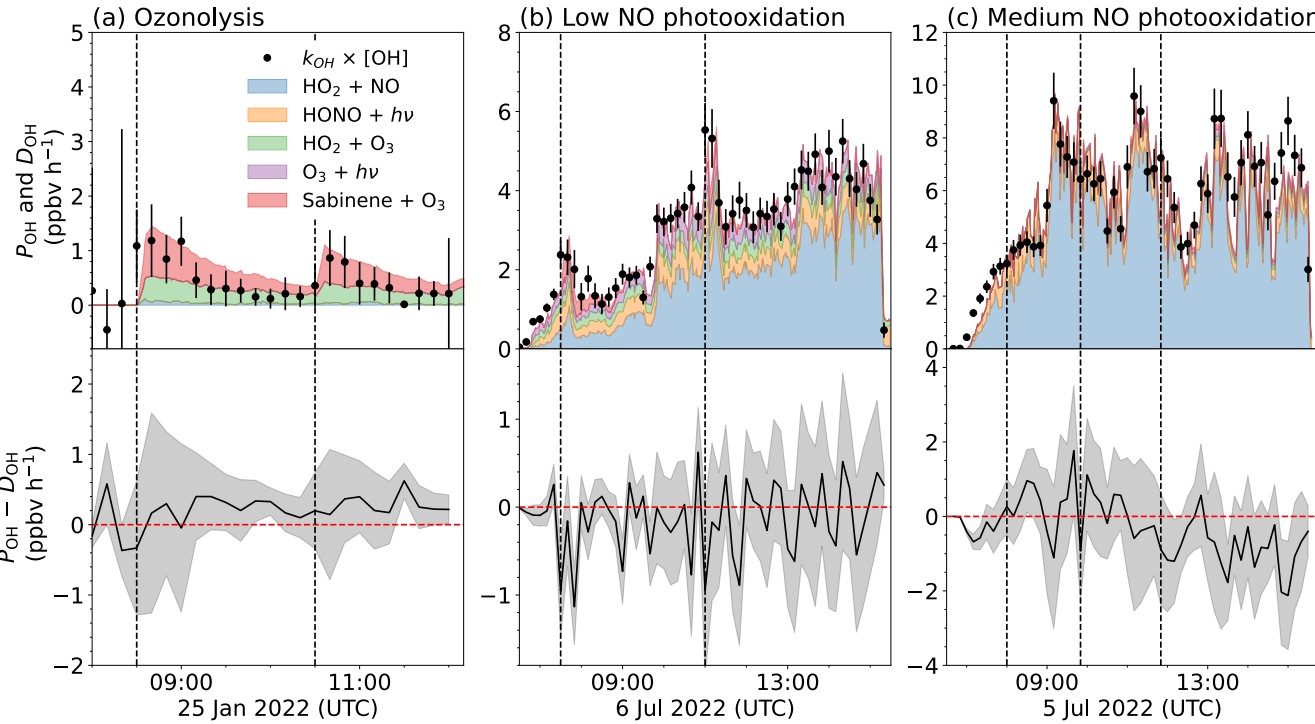

**Figure 8.** Overview of the production rate of OH ($P_{OH}$, coloured areas) and the destruction rate of OH ($D_{OH}$, black dots) (top panel), as well as their differences (bottom panel) in (a) an ozonolysis experiment, (b) experiments with low NO, and (c) with

medium NO mixing ratios. Grey shaded areas are the 1σ uncertainty of the difference between $P_{OH}$ and $D_{OH}$.

## 5 Conclusions

The oxidation of sabinene by OH and $O_3$ was investigated at different levels of NO ranging from zero to 2 ppbv in experiments conducted in the atmospheric simulation chamber SAPHIR. The experiments were performed at sabinene mixing ratios of 3

ppbv to 8 ppbv with $O_3$ mixing ratios ranging from zero to 200 ppbv.

In addition to the chamber experiments, the Arrhenius expression of the rate constant of the oxidation of sabinene by OH radicals, $k_{SAB+OH}$, was determined from measurements using an OH reactivity instrument for temperatures ranging from 284 K to 340 K, giving $(1.67\pm0.15)\times10^{-11}\times\exp((537\pm30)T^{-1})$ cm$^3$ s$^{-1}$. This agrees with the value determined in the chamber



experiments at room temperature and the experimental value reported by Atkinson et al. (1990) within the uncertainties. The
temperature-dependence coefficient of the Arrhenius expression of the rate constant $k_{SAB+OH}$ is similar to those of the OH-
oxidation of structurally-similar β-pinene and isobutene within uncertainties (Mellouki et al., 2021). The ozonolysis reaction
rate constant ($k_{SAB+O3}$) determined in the chamber experiment at 278 K is $(3.7\pm0.5)\times10^{-17}$ cm$^3$ s$^{-1}$, which is 55 % lower than
the value determined in Atkinson et al. (1990) at 298 K.

Yields of oxidation products from the reaction of sabinene with OH and O$_3$ including HCHO, acetone, and sabinaketone were
quantified in the chamber experiments. All yields determined in this study agree with yields reported in previous laboratory
studies (e.g., Reissell et al., 1999; Aschmann et al., 2002; Carrasco et al., 2006; Chiappini et al., 2006).

The product yields calculated from the experiments in this study is compared to the mechanism proposed by Wang and Wang
(2018). For the oxidation of sabinene by OH, the HCHO yield confirms branching ratios of the OH-addition reaction forming
the RO$_2$ radical SABINOHAO2, but the sabinaketone yield is lower than that branching ratio.

The experimental acetone yield is 15 % absolute higher than the value expected from the mechanism, which could be explained
by a fast isomerization reaction of the RO$_2$ radical SABINOHBO2 in the mechanism outcompeting its bimolecular reaction
with NO that produces acetone. The experimental acetone yields under NO mixing ratios between 0.1 ppbv to 0.5 ppbv are
similar. These findings suggest that the isomerization rate constant of the RO$_2$ radical SABINOHBO2 is slower than what
calculated by Wang and Wang (2018), or that the production of acetone is not affected by the competition between the
isomerization reaction and the bimolecular reaction with NO.

Regarding the product from the ozonolysis of sabinene, HCHO, sabinaketone and OH yields determined from the experiments
in this work are all lower than expected from the mechanism by Wang and Wang (2017). Further experiments are required to
investigate the reason for these discrepancies, though the observed differences could be explained by the large uncertainties in
the measurements.

The destruction rates of OH are in excellent agreement with the production rates of OH without considering additional
production from for example potential isomerization reactions of RO$_2$ radical derived from the oxidation of sabinene. This is
consistent with the proposed mechanism from Wang and Wang (2017), who calculated that there is no significant OH
production from isomerization reactions of RO$_2$ in the oxidation mechanism of sabinene.

*Data availability.* Data from the experiments in the SAPHIR chamber used in this work are available on the EUROCHAMP
data home page: https://doi.org/10.25326/W4QV-KY95 (Novelli et al., 2023a); https://doi.org/10.25326/8BA6-MM58
(Novelli et al., 2023b); https://doi.org/10.25326/QDQE-8Q79 (Novelli et al., 2023c); https://doi.org/10.25326/25V8-PA77
(Novelli et al., 2023d); https://doi.org/10.25326/FCYS-Y288 (Novelli et al., 2023e); https://doi.org/10.25326/B5VV-K378
(Novelli et al., 2023f); https://doi.org/10.25326/BQVM-QR53 (Novelli et al., 2023g).




*Author contributions*.

JYSP, FB, PTMC, and HF wrote the manuscript; JYSP, PTMC, and HF designed and led the chamber experiments; FB, GG, and RD set up and conducted the measurement of OH reactivity and reaction rate constant in the laboratory. BB (radiation), MF and AN (radicals and OH reactivity), PTMC (formaldehyde and OH radicals), SW and GG (organic compounds), FR

(nitrogen oxides and ozone) were responsible for the measurements of chamber experiments.

All the co-authors commented on and discussed the manuscript and contributed to the writing of the manuscript.

*Competing interests*. The authors declare to have no competing interests.

*Financial support*. This project has received funding from the European Research Council (ERC) under the European Union's Horizon 2020 research and innovation programme (SARLEP grant agreement No. 681529) and from the European Commission (EC) under the European Union's Horizon 2020 research and innovation programme (Eurochamp 2020 grant agreement No. 730997).

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
