# Peer review of "Atmospheric photooxidation and ozonolysis of sabinene: Reaction rate coefficients, product yields and chemical budget of radicals"

_EGUsphere, 2023_

## Community Comment (CC1)

Comments on egusphere-2023-1317 (ACP), "Atmospheric photooxidation and ozonolysis of sabinene: Reaction rate constants, product yields and chemical budget of radicals", by Pang et al.

This manuscript reports an experimental study on oxidation of sabinene initiated by OH radical and by $O_3$. The comprehensive comparison on yields of products between experimental measurements here and previous theoretical calculations suggested some possible new reaction pathways such as the possible diradical pathway in ozonolysis reactions. Publication is recommended.

Comments:

1. From Eq. 4 to Eq. 5, the concentrations of O3 and OH are assumed constant. This is not a proper assumption. Fig 3 and Fig 4 showed the changes of concentrations. Because the concentrations of OH and O3 are available from experimental measurements, it is probably better to simply integrate the concentrations as,

$$\frac{d[\text{SAB}]}{[\text{SAB}]} = -(k_{\text{O3}}[\text{O}_3]_t + k_{\text{OH}}[\text{OH}]_t + k_{\text{dil}})dt$$

$$\ln\frac{[\text{SAB}]_0}{[\text{SAB}]_t} = k_{\text{O3}}\int_0^t [\text{O}_3]_{t'}dt' + k_{\text{OH}}\int_0^t [\text{OH}]_{t'}dt' + \int_0^t k_{\text{dil}}dt'$$

Each integrate can be obtained from the experimental concentrations. A linear regression can be used to obtain $k_{\text{O3}}$ and $k_{\text{OH}}$, and contributions of OH and O3 in degradation of sabinene can be obtained.

2. Table 5 Yields of Products: For reaction with OH, the yields from the theoretical study (Wang & Wang, 2018) assumed 100% yield of RO from reactions of ROO + NO. This is probably not correct. There should be a fraction of RONO2 formation.

3. It should be noticed that the theoretical calculations all the time carry uncertainty. The predicted rate coefficients usually have uncertainty, about one order of magnitude when using ROCBS-QB3 energies. The results from theoretical calculations can be considered as the most probable values, or sometimes are biased. Therefore, experimentalists need to be aware of this, and theoreticians need to adjust the barrier heights to match the "reliable" experimental data, which are unfortunately rather limited and not available for most of time. Overall, I would consider reasonable agreement between the experimental measurements here and the previous theoretical calculations.

---

## Author Comment (AC1)

**Response to Community Comment CC1:**

We thank the reviewer for the valuable comments and suggestions. The input has improved the manuscript. Comments from the authors are in *italic*. Modifications in the text are marked in blue.

**Comment 1:**

*From Eq. 4 to Eq. 5, the concentrations of O3 and OH are assumed constant. This is not a proper assumption. Fig 3 and Fig 4 showed the changes of concentrations. Because the concentrations of OH and O3 are available from experimental measurements, it is probably better to simply integrate the concentrations as,*

$$\frac{d\,[SAB]}{[SAB]} = -(k_{O3}[O_3]_t + k_{OH}[OH]_t + k_{dil})dt$$

$$ln\frac{[SAB]_0}{[SAB]_t} = k_{O3}\int_0^t [O_3]_{t'}dt' + \int_0^t [OH]_{t'}dt' + \int_0^t k_{dil}dt'$$

*Each integrate can be obtained from the experimental concentrations. A linear regression can be used to obtain kO3 and kOH, and contributions of OH and O3 in degradation of sabinene can be obtained.*

**Response from authors for comment 1:**

We thank the reviewer for the suggestion. The ozonolysis rate constant recalculated with the integration method resulted in value of $(3.4\pm0.8)\times10^{-17}$ cm$^3$ s$^{-1}$, which is similar to the value obtained from the regression method $(3.7\pm0.5)\times10^{-17}$ cm$^3$ s$^{-1}$.

However, the integration method is less robust than the regression method as it is more sensitive to the choice of reference sabinene concentrations $[SAB]_0$. For this reason and since the ozonolysis coefficients obtained agree within 9 %, we still prefer to use the regression method to calculate the OH yield of the ozonolysis reaction. As the time interval used for the calculation of the OH yield is short, the error of replacing the integrated O$_3$ concentration with its mean value will be smaller compared to the calculation of the rate coefficient.

The value of rate constant $k_{SAB+OH}$ in the ozonolysis experiment (at a temperature of ~ 278K) is not calculated as the loss of sabinene was mostly from the ozonolysis reaction, therefore the uncertainty of $k_{SAB+O3}$ would lead to a very large uncertainty in $k_{SAB+OH}$. Values of the rate constant $k_{SAB+OH}$ in the photooxidation experiments (at around 300K) were determined using a simplified chemical model, which is very similar to the integration method. The value obtained was then compared with the rate determined from the OH reactivity measurement in the laboratory.

The manuscript is modified accordingly:

L327-369:

"Solving the differential equation Eq. (3) yields the following expression:

$$\ln \frac{[SAB]_0}{[SAB]_t} = k_{SAB+O_3} \int_0^t [O_3]_{t'} dt' + k_{SAB+OH} \int_0^t [OH]_{t'} dt' + \int_0^t k_{dil} dt' \qquad (1)$$

The reaction rate coefficient $k_{SAB+O3}$ is determined …. Using Eq. (7) to determine the OH yield by obtaining the regression coefficient $k_{loss}$ is more robust than using Eq. (1) and calculates the OH yield for every timestep within the time interval of analysis, as regression is less sensitive to the choice of reference sabinene concentration $[SAB]_0$."

L372-376:

"The rate coefficient $k_{SAB+OH}$ was determined by minimizing the root-mean-square error between sabinene concentrations measured by the PTR-TOF-MS instrument and calculations using a simplified chemical model as described in Hantschke et al. (2021). The chemical model calculates the loss rate of sabinene with measured dilution rate, OH and $O_3$ concentrations with a time step of 1 minute. The simplified model only includes the chemical loss of sabinene by the reactions with OH and $O_3$ and by dilution, without other secondary chemistry."

Values of $k_{SAB+O3}$ changed from $(3.7 \pm 0.5) \times 10^{-17}$ cm$^3$ s$^{-1}$ to $(3.4 \pm 0.8) \times 10^{-17}$ cm$^3$ s$^{-1}$ in

L16, L404, L630, Table 3, Table 4

**Comment 2:**

*Table 5 Yields of Products: For reaction with OH, the yields from the theoretical study (Wang & Wang, 2018) assumed 100% yield of RO from reactions of ROO + NO. This is probably not correct. There should be a fraction of RONO2 formation.*

**Response from authors for comment 2:**

The production of organic nitrates was overlooked in the calculation, the expected yield of formaldehyde and acetone should be lower than the values stated in Table 5. Using organic nitrate yields of (20 – 35) % usually applied for $RO_2$ radicals derived from the oxidation of monoterpenes (Rollins et al., 2010), and the OH-additional branching ratio of 47 %, we expect that the yields for HCHO and sabinaketone were about (31 – 38) %.

We also overlooked the unimolecular reaction for peroxy radical SABINOHBO2 when comparing the expected acetone yield based on Wang and Wang and the yield determined in the experiment. The expected acetone yield of 45 % stated on the manuscript only considers the branching ratio of the OH-

addition reaction, which does not consider the production of organic nitrate as mentioned, as well as the unimolecular reaction of peroxy radical SABINOHBO2 and the fraction of SABINOHBO2 that undergoes bimolecular reactions. The expected acetone yield after taking account into the unimolecular reaction rate for peroxy radical SABINOHBO2 of 5 s$^{-1}$ in the photooxidation experiments should be around 4 %, which is lower than the experimental yield of about 20 %.

We removed the yield values derived from theoretical calculations presented in Table 5.

The manuscript is modified accordingly.

L506-512:

"In the OH oxidation mechanism by Wang and Wang (2018), HCHO is only produced from the subsequent chemistry of the RO$_2$ radical SABINOHAO2 that results from one of the two OH-addition reactions of sabinene (reaction pathway (a), Fig. 1). It is reasonable to expect that about 20 % to 35 % of RO$_2$ derived from the OH-oxidation of sabinene forms organic nitrates when it reacts with NO based on the study of peroxy radicals derived from monoterpenes oxidation (e.g., Rollins et al., 2010), Therefore, the HCHO expected from the OH-oxidation of sabinene should be 31 % to 38 % when considering the branching ratio of reaction pathway (a) stated in Wang and Wang (2018) and the organic nitrate yield. This agrees with the HCHO yield of (46±25) % (Table 5) determined in the photooxidation experiments."

L515-519

" … SABINOHBO2 that undergoes an isomerization reaction or forms organic nitrates in the reaction with NO, from which eventually acetone is produced. For atmospheric conditions like in the experiments in this work, it is expected that more than 90 % of SABINOHBO2 undergoes the unimolecular reaction ($k_{uni} \sim 5$ s$^{-1}$, Fig. 1) and less than 10 % undergoes bimolecular reactions ($k_{RO2} < 0.4$ s$^{-1}$, Fig. 4 and Figs. S2 and S3). Therefore, the acetone yield is expected to be only around 4 % (reaction path (b) in Fig. 1)."

L520

"The experimentally determined acetone yield of (21±15) % is significantly higher than this value."

L535-537

"The sabinaketone yield of (19±16) % from the OH-oxidation of sabinene determined from the experiment is lower than the yield of 31 % to 38 % expected from the mechanism by Wang and Wang (2018) after taking account into the branching ratio of reaction pathway (b) in Fig. 1 and the potential production of organic nitrates."

**Comment 3:**

*It should be noticed that the theoretical calculations all the time carry uncertainty. The predicted rate coefficients usually have uncertainty, about one order of magnitude when using ROCBS-QB3 energies. The results from theoretical calculations can be considered as the most probable values, or sometimes are biased. Therefore, experimentalists need to be aware of this, and theoreticians need to adjust the barrier heights to match the "reliable" experimental data, which are unfortunately rather limited and not available for most of time. Overall, I would consider reasonable agreement between the experimental measurements here and the previous theoretical calculations.*

**Response from authors for comment 3:**

We thank the reviewer for pointing out the uncertainties in the theoretical calculations. We assume the rate coefficient referred to here is the unimolecular rate constant for the $RO_2$ radical SABINOHBO2 that can affect the acetone yield.

The manuscript is modified accordingly.

L520-534:

"The experimentally determined acetone yield of $(21\pm15)$ % is significantly higher than this value. To bring both values into agreement, the rate constant of the unimolecular reaction of SABINOHBO2 would need to be in the same order of magnitude as the loss rates constant of bi-molecular $RO_2$ reactions in this study ($\sim 0.2$ s$^{-1}$), so that about half of the $RO_2$ radical SABINOHBO2 undergoes bi-molecular reactions (mainly with NO). The uncertainty of the unimolecular reaction rate coefficient calculated in Wang and Wang (2018) of about 5 s$^{-1}$ has an uncertainty of about one order of magnitude. Therefore, the unimolecular reaction rate constant calculated by Wang and Wang (2018) agrees with the rate constant required to reach a good agreement of acetone yield, despite the large difference between the acetone yield determined in the experiments and the yield expected from the mechanism in Wang and Wang (2018).

It should be noted that acetone might be produced from pathways other than the reaction pathways of $RO_2$ radical SABINOHBO2 as suggested in Wang and Wang (2018). This would be consistent with the observation that the acetone yield found in this study and in Carrasco et al. (2006) does not strongly depend on the NO concentration (Table 5), which would be expected, if acetone is produced from reaction pathway without other competitions. For example, acetone might be produced from the bimolecular reactions of a $RO_2$ radical that does not have competing isomerization reactions, or could be produced from a very fast isomerization reaction that bimolecular reactions cannot compete with at typical NO concentrations in the atmosphere."

---

## Author Comment (AC2)

Response to Referee Comment RC1:

Comments from the authors are in *italic*. Modifications in the text are marked in blue.

**General comments:**

*The authors have provided a comprehensive experimental study of the reactions of sabinene with OH and O₃. The novel contributions include measuring the temperature dependence of the OH reaction and analyzing the chemical budget for OH radicals. The authors take pains to present previous work, describe their experimental and data analysis methods, and compare their results with past experimental and theoretical values. Based on the scientific significance and quality, I recommend accepting the manuscript.*

We thank the reviewer for the valuable comments. They help to improve the manuscript and enrich the discussion of the ozonolysis mechanism.

**Specific comments:**

*The authors may want to consider other ozonolysis reaction pathways that may affect OH yield. This includes decomposition of chemically activated CH₂OO that produces OH (see Pfeifle et al., J. Chem. Phys. **2018**, 148, 174306) and rearrangments of vinyl hydroperoxides that may reduce OH yield (see Barber et al., J. Am. Chem. Soc. **2018**, 140, 10866-10880).*

**Authors Response:**

Additional discussion on the reaction pathway that may affect the OH yield is now included, which includes the small additional OH production (3 %) from the dissociation of CH₂OO. The rearrangement of vinyl hydroperoxides could be an explanation to reduce the OH yield expected from the mechanism. Figure 8 is added to illustrate the additional pathway of vinyl hydroperoxides.

L562-579:

"The OH yield from the ozonolysis reaction of sabinene of (26±29) % determined in the experiments is lower than the yield of 44 % expected from the mechanism in Wang and Wang (2017), but still within agreement due to the uncertainties presented in the experiment and the theoretical calculation. The OH yield is derived from the fraction of Criegee intermediate CI-1 that undergoes unimolecular decomposition forming an OH radical and an β-oxo alkyl radical (Fig. 2). … The OH radical could reorientate and then recombine with the β-oxo alkyl radical, which would result in the production of 2-hydroxylketone (Barber et al., 2018; Kuwata et al., 2018, Fig. 8). With this additional mechanism, the theoretical OH yield of the ozonolysis of sabinene can be reduced. Though further investigation on the relative importance of the recombination pathway to the dissociation pathway is needed."

*The authors mention a signficant discrepancy between their measured sabinene + O₃ rate constant and the SAR value from Jenkin. Some additional discussion of the possible origin of this discrepancy would be illuminating.*

**Authors Response:**

Additional discussion of the possible origin of the discrepancy is now included.

L413-417:

"The large difference between the sabinene ozonolysis rate constants $k_{SAB+O3}$ determined experimentally and from the SAR developed by Jenkin et al. (2020) is likely related to the ring strain of the bicyclic ring. Species in that SAR with ozonolysis rate constants differing by more than a factor of three are mostly polycyclic compounds (e.g., camphene, α-copaene, and 3-carene) including sabinene. Since the SAR was constructed mostly with acyclic and monocyclic alkenes, it is likely that impacts of ring strain on the ozonolysis rate constant for polycyclic species cannot be properly captured."

**Technical corrections:**

*It would be helpful if the authors briefly summarized the theoretical methods employed by Wang and Wang. This would give the reader some sense of how reliable the theoretical predictions are without having to look up the references.*

**Authors Response:**

Brief descriptions of the theoretical method employed by Wang and Wang are now included in the manuscript.

L72-77:

"In the theoretical calculations conducted by Wang and Wang (2017 and 2018), molecular structures were first optimized and the vibrational frequencies were calculated at M06-2X/6-311++G(2df,2p) level. Electronic energies were calculated by wave function (UCBS-QB3) for the OH-oxidation of sabinene and at the RHF-UCCSD(T)-F12a level for the sabinene ozonolysis reaction. High-pressure limit rate constants were determined using the canonical transition state theory, whereas fast unimolecular reactions and their dependence on pressure and temperature were calculated with master equations (RRKM-ME) using Mesmer and MultiWell-2017 codes."

*There may be a few too many details presented in Section 3.3.*

**Authors Response:**

Details regarding to the measurement of $O_3$, NO, $NO_2$, and photolysis frequencies are now removed from the text.

L194-197

"Concentrations of trace gas ($O_3$, NO, $NO_2$, VOCs) and radical species (OH, $HO_2$, $RO_2$), photolysis frequencies (Bohn et al., 2005; Bohn and Zilken, 2005), and OH reactivity were measured in the chamber experiments in this work.

 The set of instruments used is listed in Table 1. Only the descriptions of measurements of species and quantity of interest (radicals, OH reactivity, and VOCs) in this study are included below."

*The LIF instrument, since its 1σ uncertainty is smaller than that of the DOAS instrument, has the **higher** precision (contrary to line 207). Along similar lines, the relevant column heading in Table 1 is better termed **uncertainty** than **precision**.*

**Authors Response:**

Sentence 207 is now corrected that the LIF instrument has a higher precision than the DOAS instrument.

We prefer to use the term 'precision' to describe the fluctuation of measured values (i.e., statistical error), as 'uncertainty' was already used to describe the potential systematic error caused by the method of analysis (e.g., measurement of sabineketone in L247)

L219-221

"OH concentrations measured by the LIF instrument were used for the analysis of the ozonolysis experiments due to the higher time resolution and precision compared to that by the DOAS instrument."

*There are a few typographical errors to correct.*

**Authors Response:**

Several typographical errors were found and corrected. The corrected typos were underlined.

L241-242:

"DOAS measurements were used for the analysis on that day as the CRDS method requires correction factors, whereas the DOAS method directly gives concentration values (Glowania et al., 2021)."

L22-23:

"In the ozonolysis experiments, the analysis of product measurements results in an acetone yield of (5±2) %, a formaldehyde yield of (48±15) %, a sabinaketone yield of (31±15) %, and an OH radical yield of (26±29) %."

L26-28:

"The analysis reveals that the destruction rate of OH radical matches the production rate of OH suggesting that there is no significant missing OH source for example from isomerization reactions of peroxy radicals for the experimental conditions in this work."

L134-137:

"In a temperature-controlled flow tube, OH radicals are generated in situ by photolysis of $O_3$ using laser pulses of a quadrupled Nd:YAG laser at a wavelength of 266 nm and a low pulse repetition rate of 1 Hz. $O(^1D)$ atoms produced from the photolysis of $O_3$ react with water vapor present in the gas mixture to produce OH radicals. Air containing a well-known concentration of sabinene is continuously passed through the flow tube."

---

## Author Comment (AC3)

Response to Referee Comment RC2:

Comments from the authors are in *italic*. Modifications in the text are marked in blue.

We thank the reviewer for the valuable comments. They help to improve the manuscript.

**Referee Comment 1 – 2:**

- *Rate coefficient preferred over rate constant, although the latter is used in common parlance.*
- *Bimolecular rather than Bi-Molecular (and elsewhere), you have unimolecular already without a hyphen.*

**Authors Response 1 – 2:**

The text was modified accordingly, 'rate constant' is replaced with 'rate coefficient' and 'bi-molecular' is replaced with 'bimolecular'.

**Referee Comment 3:**

*Are there plans to use a model to calculate the radical levels, or products? Will a mechanism for sabinene oxidation be developed to extend for example the MCM?*

**Authors Response 3:**

In the future it might be possible to develop a complete oxidation mechanism for sabinene including the formation of radicals.

At this stage of the study, however, the information from our experiments for specific product species is not complete enough to compare our results to model simulations. For example, $RO_2$ measurements in our study only give the total $RO_2$ concentrations but not speciated RO2 concentrations.

**Referee Comment 4 – 9:**

- *Line (L) 13. exp((575±30)T-1) would be better as exp((575±30)/T), also on L390, and elsewhere (e.g. Table 4).*

- *... the analysis of (not if)*

- *O3 (Wang and Wang, 2017), ..... add respectively*

- *temperatures rather than temperature*

- *In situ and not in-situ*

**Authors Response 4 – 9:**

The text was modified accordingly following the order above in

- L13, L420, L431 (Table 4), L631
- L22
- L71
- L132
- L134

**Referee Comment 10:**

*L128, it is probably worth adding followed by reaction of the O(1D) atoms generated with water vapour present in the gas mixture.*

**Authors Response 10:**

The text was modified accordingly.

L136: "In a temperature-controlled flow tube … $O(^1D)$ atoms produced from the photolysis of $O_3$ react with water vapor present in the gas mixture to produce OH radicals."

**Referee Comment 11:**

*L138 Sabinene is measured using a TOC instrument with detection of the CO2 formed after pyrolysis. Is there good evidence that 100% of the sabinene is removed to form CO2? Also, is the reason why a TOC is used for the high time-resolution required? It may be that the time-resolution is not good enough for the experiments for other methods (e.g. MS) – but perhaps a note to note this is needed.*

**Authors Response 11:**

We have good evidence that sabinene was quantitatively converted to $CO_2$, as this method was also applied to other VOCs (alkanes, aromatics, monoterpenes) to measure their OH-reaction rate coefficients. Rate coefficients of the OH-oxidation reaction of these VOCs agree well with values recommended by IUPAC, suggesting that the VOC concentrations were accurately measured by the TOC method. Therefore, we are confident in the quantification of the VOCs concentration in the canister using the TOC method. Additional description is included in the text:

L149-153: "The catalytic conversion from VOCs to $CO_2$ was tested with other VOCs (alkanes, aromatics and monoterpenes) and showed a complete conversion. Therefore, it can be assumed that sabinene was completely converted to $CO_2$ during the TOC measurement. Assuming that all carbon stems from sabinene, its concentration in the canister can be calculated from the measured $CO_2$ concentrations."

A measurement with a GC would have been also an option to determine the sabinene concentration in the experiment. However, the accuracy of the $CO_2$ detection by the CRDS instrument is higher than that of sabinene detection by the GC instrument as there was no GC calibration standard for sabinene available.

**Referee Comment 12:**

*L144 and L148. Temperatures here are in Celsius, whereas in the abstract they are given in K. I suggest that the abstract retains K as given, but perhaps K in brackets after the T in Celsius might be useful to add. Otherwise it will be confusing as for equation (2) on line 158 the T here has to be in K.*

**Authors Response 12:**

The text was modified accordingly in

L146-148: "In this method, a small flow (500 sccm) from the canister flowed through a pre-oven at 760 °C (1033 K) and afterward over a palladium catalyst at 500 °C (773 K)."

L155 (283 K to 343 K), L160 (283 K to 343 K), and L171 (0.005 % at 293 K).

**Referee Comment 13:**

*Please state the zero reactivity of the OH reactivity instrument.*

**Authors Response 13:**

The zero reactivity was about 2 $s^{-1}$ to 3 $s^{-1}$. The text was modified accordingly:

L163-166: "The OH reactivity of air with sabinene was subtracted by its corresponding zero reactivity ranging from 2 $s^{-1}$ to 3 $s^{-1}$. The rate coefficient of the OH reaction was calculated by using the sabinene concentration in the canister ($[SAB]_0$) and the dilution factor $f_{dil}$ determined from the flow rates:"

**Referee Comment 14 – 15:**

- Replace -EAR-1 by $-E_A/R$ to be consistent with the equation

- Please provide a reference for the ROxLIF instrument.

**Authors Response 14 – 15:**

The text was modified accordingly in

- L173
- L212 (Fuchs et al., 2011)

**Referee Comment 16:**

*L200-201. Is the HOx cell the same as the second LIF cell where HO2 is measured? Is the ROx cell the same as "another low-pressure LIF detection cell". Make this clear if the case. as HOx and ROx cells are not defined at present*

**Authors Response 16:**

The $HO_x$ cell is the same as the LIF cell where $HO_2$ is measured. The text was modified accordingly:

L202-215 "The LIF instrument consists of three measurement cells for the separate detection of OH, $HO_2$, and $RO_2$ radicals … Similarly, the $RO_2$ concentration is finally derived from the difference between the fluorescence signals obtained in the $RO_x$ and the $HO_x$ cells."

**Referee Comment 17:**

*L202-207. It is excellent that there are two independent in situ methods for measuring OH. This is unique to SAPHIR. However, the statement that the DOAS and LIF instruments agreed with each other is a bit confusing if the difference between the 2 instrument is similar to the concentration of OH during the experiments? This suggests a significant difference? I agree though that because the OH concentration is similar to or below the 1-sigma precision of the DOAS instrument, that actually only the LIF instrument value is used.*

**Authors Response 17:**

We think that the agreement between OH concentrations measured by the DOAS and LIF instruments in the ozonolysis experiments is good as the absolute difference between the two values is small. However, if OH concentrations are close to the limit of detection of the instrument, the relative difference can become large. The text was rephrased as follow:

L216-221: "In three of the experiments, both the LIF and DOAS instruments were available. In the two ozonolysis experiments on 24 and 25 January 2022 (Table 5), mean OH concentrations measured by the DOAS and LIF instruments were both low at around 0 to $1 \times 10^6$ cm$^{-3}$ (Fig. 3 and Fig. S1). The mean value of the difference between OH concentrations measured by the two instruments was about $0.7 \times 10^6$ cm$^{-3}$. This is close to the limit of detection of the DOAS instrument (**Error! Reference source not found.**). OH

concentrations measured by the LIF instrument were used for the analysis of the ozonolysis experiments due to the higher time resolution and precision compared to that by the DOAS instrument."

**Referee Comment 18:**

*L212-213. DOAS is used for the OH measurements in the photooxidation experiments, and there is a note that the difference with LIF might be due to an unaccounted calibration error. Is there confidence then in the LIF value for the O3 experiments when only the LIF value is used?*

**Authors Response 18:**

For the calculations of kinetic parameters from the ozonolysis experiment, we did not make use of OH concentration measurements. OH measurements by the LIF instrument in the ozonolysis experiment were only used to calculate the OH production rate during the experiment (Fig. 9). The overall small production rate of less than 1 ppbv/h and the low precision of OH measurements give a scatter of data points of approximately 50%. An additional uncertainty of 25% in the OH concentration values, would therefore not change any of the conclusions.

**Referee Comment 19:**

*Can a note be added about why the PTR was not calibrated for sabinene. Is there an experimental limitation which prevents this?*

**Authors Response 19:**

Unfortunately, there was no calibration standard available to accurately determine the sensitivity of the PTR-MS instrument for sabinene. The scaling of the ion mass signal to match the increase in the measured OH reactivity at the point of injection of a VOC has been proven to give an accurate estimate of the calibration factor of PTR instruments in previous chamber experiments if the OH reaction rate constant is known. In this work, the OH reaction rate constant was independently measured and therefore well known.

**Referee Comment 20:**

*factors not factorss*

**Authors Response 20:**

The text was modified accordingly in L242.

**Referee Comment 21:**

*There are quite a few Novelli et al., 2023 references (a, b, c, ....), and specific figures are referred to in these (which are Eurochamp datasets), e.g. from Table 2 and also in the text. The figures were all blank for me when I clicked on the links (the axes show but no line or point on the graphs). I wonder if the Supplementary Information (SI) for this paper could be used to display these specific figures instead of having to click on links (which did not any data for me) – it would be easier for the reader? I think if the intention of the references is to contain significant supporting data for each experiment, then this is fine (but the figures were blank for me), but if specific Figures from these databases are cited, then these ought to be more readily observable via the SI.*

**Authors Response 21:**

Time series of all trace gases and radicals from experiments in this work can be found in the figures of the main paper and in supplementary information. The reference to the EUROCHAMP database is only meant to link to the data but is not thought to give additional information. Unfortunately, there is indeed some problems with displaying time series properly on the webpage of the EUROCHAMP database.

**Referee Comment 22:**

*Figure 3. For the last panel, the loss of RO2 seems to be completely dominated by RO2+HO2, with RO2+RO2 being very small indeed. Given that the concentrations of RO2 and HO2 are similar (two other panels), then I think a comment is needed to say that the RO2+RO2 rate coefficient is 20 times smaller than for RO2+HO2 (this information is in the SI).*

**Authors Response 22:**

Additional information was added to the text to describe the fraction of $RO_2$ undergoing different bimolecular reaction pathways.

L263-266: "$RO_2$ radicals were expected to react exclusively with $HO_2$ radicals in the absence of NO, if only bimolecular reactions of $RO_2$ radicals are considered. The self-reaction between $RO_2$ radicals is expected to be of minor importance compared to the reaction with $HO_2$ radicals, as the reaction rate constant of self-reactions of $RO_2$ is about 20 times slower than that of the reaction with $HO_2$ radicals (Supplementary material Section 1)."

L283-286: "Over 80 % of $RO_2$ radicals were expected to react with NO and the remaining part mostly reacted with $HO_2$ radicals in the photooxidation experiments, if only bimolecular reactions of $RO_2$ radicals are considered. For the experiment on 06 July 2022, only around 60 % $RO_2$ radicals reacted with NO at the beginning due to the low NO concentration caused by cloudy weather."

**Referee Comment 23 – 26:**

- *Add (100 ppmv) after CO to make this clearer again.*
- *The MCM is used to calculate photoylsis rates of ketones – can a reference please be given for this.*

- *Known and not know (~ L407)*

- *"OH production rate was excellently balanced", better might be ".... " the OH production was very well balanced..."*

**Authors Response 23 – 26:**

The text was modified accordingly in

- L294
- L321 (Atkinson et al. 2006)
- L407
- L608